

# A 125-year record of climate and chemistry variability at the Pine Island Glacier ice divide, Antarctica

Franciele Schwanck[1], Jefferson C. Simões[1], Michael Handley[2], Paul A. Mayewski[2], Jeffrey D. Auger[2], Ronaldo T. Bernardo[1], Francisco E. Aquino[1]

[1]Centro Polar e Climático, Universidade Federal do Rio Grande do Sul (UFRGS), Porto Alegre, 91540-000, Brazil
[2]Climate Change Institute, University of Maine, Orono, 04469, United States

*Correspondence to*: Franciele Schwanck (franschwanck@gmail.com)

**Abstract.** The Mount Johns (MJ) ice core (79º55'S; 94º23'W) was drilled near the Pine Island Glacier ice divide on the West Antarctic Ice Sheet during the 2008–2009 austral summer, to a depth of 92.26 m. The upper 45 m of the record covers

approximately 125 years (1883- 2008) showing marked seasonal variability. Trace element concentrations in 2,137 samples were determined using inductively coupled plasma mass spectrometry. In this study, we reconstruct mineral dust and sea salt aerosol transport and investigate the influence of climate variables on the elemental concentrations to the MJ site. The ice core record reflects changes in emissions as well as atmospheric circulation and transport processes. Our trajectory analysis shows distinct seasonality, with strong westerly transport in the winter months and a secondary northeasterly transport in the

summer. During summer months, the trajectories present slow-moving (short) transport and are more locally influenced than in other seasons. Finally, our reanalysis trace element correlations suggest that marine derived trace element concentrations are strongly influenced by sea ice concentration and sea surface temperature anomalies. The results show that seasonal elemental concentration maxima in sea-salt elements correlate well with the sea ice concentration winter maxima in the West Amundsen and Ross Seas. Lastly, we observed an increased concentration of marine aerosols when sea surface temperature

decreased.

## 1 Introduction

The West Antarctic Ice Sheet (WAIS) is more susceptible to marine influences than that of the East Antarctica Ice Sheet (EAIS). The lower average elevation of the WAIS compared to EAIS, 1100 m and 3000 m, respectively, (Bedmap 2 project data; Fretwell et al., 2013) facilitates the advection of air masses toward the interior of the continent, thereby directly

contributing to the ice sheet's surface mass balance through precipitation (Nicolas and Bromwich, 2011). During recent decades, rapid changes have occurred in the WAIS sector, including flow velocity acceleration, retraction of ice streams, and mass loss (Pritchard et al., 2012).

Presently, Pine Island Glacier (PIG) is responsible for 20 % of the total ice discharge from the WAIS (Rignot et al., 2008). Accelerated thinning observed since the 1980s is directly linked to enhance sub-ice-shelf melting, which is induced by the

recent alteration of Circumpolar Deep Water (Pritchard et al., 2012; Steig et al., 2012; Favier et al., 2014). Sea level pressure



and geopotential height anomalies, associated with increased strength of the circumpolar westerlies (Steig et al., 2012), favor reduced sea ice extent in the Amundsen and Bellingshausen Seas (Schneider et al., 2011) and the advection of warm air onto the continent (Steig et al., 2009; Ding et al., 2011). In this context, both atmospheric and oceanic variability are important for determining the response of the WAIS over long time-scales

Polar ice cores contain important chemical records linked to the climate that extend continuously for the past 800,000 years (Lüthi et al., 2008; Delmonte et al., 2008), making them a valuable tool for interpreting climate trends. Despite the increase in ice core sites in Antarctica, they are spatially distant. In particular, there is a lack of ice core records from the WAIS. Previous studies have emphasized the importance of measuring elemental composition of ice cores in this region in order to understand the recent atmospheric circulation changes (Criscitiello et al., 2014; Pasteris et al., 2014; Tuohy et al., 2015). To

interpret chemistry records from Antarctic ice cores, it is imperative to recognize the long-range transportation of continental dust and regionally derived sea salt, furthermore the seasonality of atmospheric loading and deposition onto the Antarctic continent.

Major and minor impurities in polar ice originate from sources such as oceans, landmasses, volcanism, biogenic activity, biomass burning, and anthropogenic inputs (Legrand and Mayewski, 1997; Planchon et al., 2002; Vallelonga et al., 2004;

Weller et al., 2008; Dixon et al., 2013; Schwanck et al., 2016). These aerosols are transported over long distances to the polar regions through the troposphere (Petit and Delmonte, 2009) and stratosphere, e. g. volcanic emissions (Krinner et al., 2010). Sea salt and mineral dust can be used to reconstruct climate conditions as well as atmospheric transport patterns (Albani et al., 2012a; Chewings et al., 2014). These aerosols are strongly influenced by the climate conditions in the source region, and the depositional record provides important information about cyclone activity, wind intensity (Koffman et al.,

2014), sea ice conditions (Criscitiello et al., 2014), and aridity and vegetation cover (McConnell et al., 2007).

Sources can be either primary aerosols, such as sea salt or mineral dust, or secondary aerosols, such as those produced in the atmosphere through oxidation of trace gases (Legrand and Mayewski, 1997). Sea salt is produced by two dominant mechanisms: 1) bubble bursting and sea spray over open water (Leeuw et al., 2011), and 2) by the formation of frost flowers on the surface of young sea ice at high latitudes (Kaspari et al., 2005; Fischer et al., 2007). Marine aerosol concentrations are

strongly linked to cyclone frequency and intensity, increasing wind speeds over the ocean surface for efficient transport then depositing aerosols along the storm track (Fischer et al., 2004). The other primary source of aerosols, mineral dust, is transported from arid, continental regions such as Australia (Li et al., 2008), South America (Delmonte et al., 2010; Li et al., 2010), and New Zealand (Neff and Bertler, 2015) to the Antarctic. Once entrained into the atmosphere, small grains (0.1 to 5 μm) can be transported over long distances (Gaiero et al., 2007; Mahowald et al., 2014) through advection before they are

deposited on the Antarctic snow surface (Prospero et al., 2002; Delmonte et al., 2013).

Here, we present a 125-year record of eleven trace elements (Al, Ba, Ca, Fe, K, Mg, Mn, Na, S, Sr, and Ti) from the Mount Johns (MJ) ice core in the WAIS. We focus on the influence of atmospheric circulation, surface temperature, and sea ice concentration on the transportation of mineral dust and sea salt aerosol to the MJ site. Correlations were made between trace element concentrations and the European Centre for Medium Range Weather Forecasts (ECMWF) Reanalysis Interim



(ERA-Interim; Dee et al., 2011) climate variables. Modeled air mass trajectories were also employed to explore atmospheric transport, independent of deposition processes, from 1979 to 2008.

## 2 Methods

### 2.1 Site description and core collection

Pine Island, Thwaites, and Smith Glaciers are the principal drainage systems into the Amundsen Sea (Shepherd et al., 2002;
Rignot et al., 2002). These glaciers receive a significant amount of precipitation due to their low elevation and coastal locations. The lower elevation allows moisture-rich cyclones to penetrate deeper into the interior (Nicolas and Bromwich, 2011). Kaspari et al. (2004) presented accumulation records from four ice core sites collected within the Pine Island-Thwaites drainage system during the International Trans-Antarctic Scientific Expedition (ITASE). Based on these records, they found that recent snow accumulation (between 1970 and 2000) had increased when compared to the 1922–1991
average. However, radar-derived annual accumulation records show no significant trend over Thwaites Glacier between 1980 and 2009 (Medley et al., 2013).

Satellite observations show ice mass loss along the Bellingshausen and Amundsen Seas has increased over, at least, the past two decades (Rignot et al., 2008) as a result of ocean-driven basal melt (Pritchard et al., 2009; Pritchard et al., 2012). WAIS is grounded below sea level where warming waters rapidly melt the ice shelves resulting in increased glacial flow (Mercer,
1978; Bamber et al., 2009), further draining the ice sheet (Steig et al., 2012; Dutrieux et al., 2014). Regional changes in atmospheric circulation and associated changes in sea surface temperature and sea ice extent also directly influence the warming trend in West Antarctica (Steig et al., 2009; Pritchard et al., 2012). Recent studies have indicated that the pronounced warming in the central tropical Pacific Ocean is related to a pressure anomaly north of the Amundsen Sea Embayment, an increase in sea surface temperatures (SST), and negative anomalies in sea ice extent (Steig et al., 2009; Ding
et al., 2011; Steig et al., 2012).

In this study, we use an ice core located near the Pine Island Glacier divide (Fig. 1) to reconstruct mineral dust and marine aerosol transport and the influence of climate variables on the elemental concentrations. The MJ ice core (79°55'28" S, 94°23'18" W, and 91.20 m depth) was recovered in the austral summer of 2008/2009. The ice thickness at the ice core site reaches 2,115 m (determined from the Bedmap 2 project data; Fretwell et al., 2013). The average accumulation rate for the
period is 0.21 m water equivalent per year (based on calculated field measurements of snow and ice density) and the mean surface temperature (measured at a depth of 12 m) is -33ºC, measured using a calibrated platinum probe.

Drilling was performed using the Fast Electromechanical Lightweight Ice Coring System (FELICS) (Ginot et al., 2002). The MJ core (8.5 cm diameter) was cut into sections of approximately 1 m in length, packed in polyethylene bags and then stored in high-density Styrofoam boxes and transported by air to Punta Arenas, Chile. Then, it was sent frozen to the Climate
Change Institute (CCI) at the University of Maine, USA, where it was sub-sampled and analyzed using Inductively Coupled Plasma Sector Field Mass Spectrometry (ICP-SFMS).



## 2.2 Laboratory analyses

Decontamination of the MJ ice core was carried out in a certified cold room (-20°C) of class 100. Prior to melting, core ends and breaks were manually scraped using a clean ceramic knife to reduce potential contamination from drilling or other sources (procedure performed according Tao et al., 2001 and described in Schwanck et al., 2016). The core was melted using a continuous ice core melter system developed by CCI researchers (details in Osterberg et al. 2006). This system uses fraction collectors to gather discrete, high resolution, continuous and co-registered meltwater samples. Our samples were collected into acid-cleaned (Optima $HNO_3$), low density polyethylene (LDPE) vials and acidified to 1 % with double-distilled $HNO_3$ under a class-100 High Efficiency Particle Air (HEPA) clean bench and allowed to react with the acid (at room temperature) at least one month before analysis. This process is important for the dissolution of particulate material (following studies Rhodes et al., 2011 and Koffman et al., 2014).

Trace element concentrations in 2,137 discrete samples, corresponding to the upper 45 m of the MJ ice core, were determined using the CCI Thermo Electron Element2 ICP-SFMS coupled to an ESI model SC-4 autosampler. Working conditions and measurement parameters are described in Table S1 (Supporting Information). The ICP-SFMS is calibrated daily with five standards that bracket the expected sample concentration range. Although there is no certified reference material for trace elements in polar snow and ice, the analyzed samples were certified with water reference material (SLRS-4, National Research Council Canada, Ottawa, Canada) to ensure the concentrations were within the certification range, confirming the accuracy of this method (details in Osterberg et al., 2006).

Samples of de-ionized water, or "blanks", were prepared, treated, and analyzed using the same method as snow samples. The Method Detection Limits (MDL) were defined as three times the standard deviation of blank samples (10 blank samples were used), which were 0.305 ng $g^{-1}$ for Al, 0.382 pg $g^{-1}$ for Ba, 0.095 ng $g^{-1}$ for Ca, 0.050 ng $g^{-1}$ for Fe, 0.058 ng $g^{-1}$ for K, 0.385 ng $g^{-1}$ for Mg, 0.936 pg $g^{-1}$ for Mn, 0.213 ng $g^{-1}$ for Na, 0.064 ng $g^{-1}$ for S, 0.741 pg $g^{-1}$ for Sr, and 0.660 pg $g^{-1}$ for Ti. Concentrations below the MDL were replaced with the MDL values, which occurred in a few cases for Al, Ba, Ca, Fe, Mn, Sr, and Ti (less than 1 %). Blank concentrations and MDLs were either similar or less than published values using comparable methods and instruments (Table S2 – Supporting Information).

Water isotope analyses were performed at the Centro Polar e Climatico, Brazil, Isotopes Lab using a Picarro L2130-i wavelength-scanned cavity ring-down spectroscopy (WS-CRDS) instrument (Picarro Inc., USA). For water isotope analysis, aliquots of water are filled in 2 ml glass vials and sealed with polytetrafluorethylene/silicone caps. The vials are then placed in a PAL COMBI-HTCxt autosampler (CTC Analytics AG, Switzerland) connected to the Picarro L2130-i for δD and δ$^{18}$O. Reproducibility of measurements is typically 0.9 ‰ for δD and 0.2 ‰ for δ$^{18}$O.

The upper 45 m of the MJ ice core encompasses the period of 1883 to 2008 (± 1 year). The ice core is dated by annual layer counting using seasonal variations of Na and S concentrations. Elevated concentrations of Na in the interior of Antarctica occur primarily during winter and spring months due to the expansion of the jet stream facilitating an increase of Rossby waves and meridional transport of air masses leading to a baroclinic state (Ding et al., 2011). Layer dating by counting is





reliable in polar ice cores as the original deposited snow sequence is preserved due to melting, percolation, and refreezing processes being rare in the Antarctic ice sheet (Cuffey and Paterson, 2010). We also identified the major volcanic eruptions during this period, such as Pinatubo (1991), Agung (1963), Santa Maria (1902), and Krakatoa (1883) (details outlined in Schwanck et al., 2016).

## 2.3 Meteorological data analyses

To explore possible sources of the observed trace elements in aerosols, air mass backward trajectories were simulated for 1000 m above ground level over the MJ ice core site. Trajectory simulations were made using the Hybrid Single-Particle Lagrangian Integrated Trajectory (HySPLIT) model, developed by the NOAA Air Resources Laboratory (Draxler et al., 2010) in conjunction with the global reanalysis datasets from the National Centers for Environmental Prediction (NCEP) and the National Center for Atmospheric Research (NCAR), known as NCEP/NCAR reanalysis model (NCEP1) (Kalnay et al.,

1996; Kistler et al., 2001). Despite limitations prior to the satellite era (1979), the NCEP/NCAR reanalysis model represents a useful tool for understanding the climate of the Southern Hemisphere since 1979 (Bromwich and Fogt, 2004) and has been used with success in Antarctic back-trajectory modeling (e.g. Sinclair et al., 2010; Dixon et al., 2011; Markle et al., 2012). Five-day (120 hr), 3-D back-trajectories were created from the MJ site at 00:00 UTC daily from January 1979 to December 2008 (a total of 10,655 trajectories). We have tested the effect of model initiation heights at 500 m, 1000 m, and 1500 m on

our trajectories and confirmed they are spatially consistent. The initial heights of 500 m are affected by surface topography, which is imperfectly represented in the reanalysis model (Dee et al., 2011). For this reason, we have chosen the 1000 m level as the initial height condition for our back trajectories. At this altitude, orographic influences are minimized while the trajectories are sufficiently close to the terrain to be dynamically linked to the surface wind field (Sinclair et al., 2010). The five day simulation is an appropriate time-length when considering the maximum lifetime transport (10 days) of small size

(0.1 – 2.5 μm) fractions of mineral dust and other aerosols, while transport of large particles (> 2.5 μm) is likely restricted to the first several days (Albani et al., 2012a). Cataldo et al. (2013) measured the size of dust particles in a separate ice core drilled near MJ and found that the mean dust size within the core ranges from 1.2 μm to 2.4 μm. In order to obtain information about airflow patterns at the MJ site, a cluster analysis was applied to a database of individual trajectories (10,655 daily trajectories). The HySPLIT model's cluster analysis algorithm groups trajectories by minimizing

the spatial variability between trajectories within some defined number of clusters (Draxler, 1999). For the trajectories presented here, it is determined that five clusters are sufficient to capture seasonal variability during the 1979 to 2008 period. Bracegirdle and Marshall (2012) determined that ERA-Interim was the most accurate of six reanalysis models over Antarctica when compared against surface and midtropospheric pressure and temperature observations. ERA-Interim was thus utilized to provide annual mean 2-m air temperature ($T_{2m}$), sea surface temperature (SST), and sea ice concentration

(SIC) from 1979 to 2008. ERA-Interim outputs were obtained from the ECMWF Data Server (http://apps.ecmwf.int/datasets/) at a resolution of 1.5º. Spatial correlations were performed between MJ ice core element data and ERA-Interim climate variables.





## 3 Results and discussion

### 3.1 Glaciochemical records

Concentrations of 11 trace elements (Al, Ba, Ca, Fe, K, Mg, Mn, Na, S, Sr, and Ti) were measured in 2,137 discrete ice core samples. Table 1 shows a statistical summary of the trace element concentrations measured from the MJ ice core. Contributions from primary natural sources in each sample were estimated using the following indicators: non-sea-salt-sulfur (nssS) for volcanic emissions, aluminum (Al) for rock and soil dust, and sodium (Na) for marine spray. Non-sea-salt ratios were calculated using Eq. (1) (Wagenbach et al., 1998):

$$nssS = (S_{total}) - k \times (Na_{ice}) , \tag{1}$$

where $k = 0.0769$ is the sea-salt ratio of sulfur to sodium corrected for sea-salt fractionation processes. This correction was calculated using a similar technique to that described by Wagenbach et al. (1998), more details in supporting information (Figure S1). In this study we used the main concentrations of the ocean water (Lide, 2005) as reference values. Approximately 40 % of the nssS in the Southern Hemisphere originates from oxidation of dimethylsulfide (DMS) produced

by marine phytoplankton during summer months (Gondwe *et al.*, 2003). This signal was removed from the record using a mean seasonal cycle, the nssS record was down-sampled into 8 even pins per year, and the 125-yr centered moving average (starting in 1883) of these bins was calculated, and then subtracted from the total nssS record (Van Ommen and Morgan, 1996).

The soil and rock dust contribution for the measured trace elements were given by the crustal Enrichment Factor (EFc)
according to Eq. (2) (Osterberg, 2007):

$$EFc = \frac{X_{ice}/Al_{ice}}{X_{ref}/Al_{ref}} , \tag{2}$$

where $X_{ice}$ is the trace element concentration in the sample, $Al_{ice}$ is the aluminum concentration in the sample, and $X_{ref}$ and $Al_{ref}$ are the trace element and the aluminum concentrations in the reference material, respectively. Aluminum was used as the reference element in this work because it is one of the major constituents of the earth's crust (Planchon et al., 2002). The
mean elemental concentration used for reference is the average composition of the upper continental crust taken from the literature (Wedepohl, 1995).

Figure 2 shows trace element EFc values during the austral summer and winter months in the MJ ice core. Elements with EFc lower than 10 are considered to be non-enriched and predominantly have a crustal dust origin (Duce et al., 1975). EFc higher than 10 indicates contributions from other sources, such as marine aerosol, volcanism, biogenic activity or
anthropogenic emissions. Marine aerosol contributions were estimated using the oceanic enrichment factor (EFo) according to Eq. (3) (Osterberg, 2007):

$$EFo = \frac{X_{ice}/Na_{ice}}{X_{ref}/Na_{ref}} , \tag{3}$$





where $X_{ice}$ is the trace element concentration in the sample, $Na_{ice}$ is the Na concentration in the sample, and $X_{ref}$ and $Na_{ref}$ are the trace element and Na concentration in the reference material, respectively. Sodium is used as the reference element

because it is the main sea salt constituent (Weller et al., 2008; Dixon et al., 2013). We used the average composition of ocean water (Lide, 2005) as a reference for the ocean elemental abundances.

Approximately 10 to 15 % of the non-sea-salt sulfate/sulfur concentration in the Antarctic atmosphere originates from volcanic activity (Boutron and Paterson, 1986; Hur et al., 2007). We used the Hinkley et al. (1999) element/S ratios to calculate inputs from the global mean volcanic quiescent degassing background for the element Mn and S (there is not data

available for the other elements). Furthermore, we used the metal/S ratios from the Mount Erebus (77°32'S, 167°10'E) plume (Zreda-Gostynska et al., 1997) to represent local source contributions for Al, Ca, Fe, K, Mn, Na, S, and Ti.

The first step to calculating the volcanic contribution is to remove the oceanic and crustal fraction of elements. These fractions were calculated according to Eq. (4) (Legrand and Mayewski, 1997):

$$X_o = Na_{ice}(X_{ref}/Na_{ref}) , \qquad (4)$$

where $X_o$ is the fraction of the oceanic origin of the element, $Na_{ice}$ is the Na concentration in the sample, and $X_{ref}$ and $Na_{ref}$ are, respectively, the trace element and Na concentrations in the reference material. The oceanic fraction was subtracted from each sample, and then, the crustal fraction was calculated using the same formula but substituting Na for Al as the reference material. We are left with the excess [excess = total– (oceanic + crustal)] elemental concentrations to calculate the local (3 % - 5 %) and global (10 % - 15 %) volcanic contributions (Table 2).

Only S and Mn show significant input of volcanic emissions with contributions ranging from 20 to 33 % from regional volcanic sources and 3 to 5 % from global emissions for mean excess elemental concentrations. The other elements (Al, Ca, Fe, K, Na, and Ti) presented less than 1 % of volcanic input. Due to lack of data in the literature, we did not calculate volcanic increments for the elements Ba, Mg, and Sr. Based on the low values of EFc and EFo, we consider that the presented concentrations are from natural origin and possible anthropogenic contributions to these elements would be

insignificant in this area.

We applied the Pearson's Correlation (Table 3) to analyze the relationship between the elements, where strong positive relationships suggest common sources or common atmospheric transport pathways (Hur et al., 2007). Al shows strong correlation with Mg (> 0.6) and moderate correlation (0.4– 0.6) with Ti, which indicate similar transport and deposition processes of these elements. Na presents a strong correlation (> 0.6) with Sr, Mn, and K and moderate correlation (0.4– 0.6)

with S, Mg, and Ba showing a relationship between these elements, possibly origin or transport common.

Based on the analysis of crustal and marine enrichment factors and Pearson correlation, we have classified the concentrations as predominantly crustal for the elements Al and Ti, while Na, Sr, and K are primarily sea salt derived elements. The elements Ba, Ca, Fe, Mg, Mn, and S show to be from mixed sources (mineral dust and sea salt aerosol). Furthermore, the S record has a considerable volcanic and biogenic input and Mn has an additional volcanic input. We acknowledge that Fe may

have an additional contribution of biomass burning aerosol (Winton et al., 2016), but due to lack of data we will not address





this issue here. A rough estimate of the contribution from dust, sea-salt, volcanoes, and biogenic activities for the MJ ice core is shown in supporting information (Table S3).

### 3.2 Seasonal aspects

Generally, trace element concentrations from sea salt aerosol observed in coastal and interior West Antarctic ice cores show
a clear seasonal signal, with higher concentrations in austral winter and spring months (June to November) and lower concentrations in austral summer months (December-February) (Legrand and Mayewski, 1997; Wolff et al., 2003; Kaspari et al., 2005; Sigl et al., 2016). Impurities from continental dust can peak in both the austral summer (Weller et al., 2008; Tuohy et al., 2015) or winter months (Hur et al., 2007), depending on site location. Additionally, biogenic aerosols (e.g. sulfur) show peaks in summer months due to an increased phytoplankton activity (Weller et al., 2011). We found high
concentrations in austral winter and low concentrations in austral summer for almost all elements analyzed, with the exception of sulfur, which presents peaks in the summer due to the biogenic contribution. This variability was confirmed by the seasonal variation of water isotopes. The $\delta^2H$ maxima represents annual summer peaks, while the $\delta^2H$ minimum marks the lowest annual temperature (associated with the winter). According to Tuohy et al. (2015), winter peaks are associated with large precipitation events and/or seasonal wind speeds. Multiple annual peaks have been observed in the Lambert
Glacier Basin area (Hur et al., 2007) and Roosevelt Island, Ross Sea (Tuohy et al., 2015). This was linked to variable atmospheric transport and aerosol loading. We observed similar events in a few years of the MJ record. The seasonal variability in the MJ ice core is exhibited in Figure 3.

Temporal variability in aerosol records generally reflects changes in atmospheric moisture transport. Spatial accumulation variability across the ice sheet is related to topography and distance from the predominant moisture source (Kreutz et al.,
2000a). West Antarctica is largely influenced by cyclonic activity that penetrates into West Antarctica, delivering heat and moisture (Kaspari et al., 2004; Dixon et al., 2012). Previous research on trace element concentrations has indicated that fresh snowfall in high-accumulation sites is generally related to the highest trace element concentrations (Wolff et al., 1998; Kreutz et al., 2000a). Although, dry deposition can contribute between 10 % and 25 % of chemical impurities in snow, and these mechanisms increase in importance with decreasing annual accumulation (Davidson et al., 1981; Tuohy et al., 2015).
The annual mean concentrations of 11 trace elements in the MJ ice core are shown in Fig. 4. Major, historical volcanic eruptions, such as Pinatubo (1991), Agung (1963), Santa Maria (1902), and Krakatoa (1883), were identified by large sulfur concentration peaks and were used as absolute time horizons during the timescale elaboration. The concentrations are highly variable down the length of the core. For example, mean Mn concentrations range from ~0.94 pg g$^{-1}$ to 450.88 pg g$^{-1}$. The highest concentrations are generally observed before 1930 for all elements. Similar trends are noted for Al, Ba, Fe, Mg, S,
and Ti. In contrast, the concentrations of K and Mn remain low (with the exception of a few peaks) and correlated well during the entire period. Ca, Na, and Sr presented low annual variability over the period but marked seasonal variability. In particular, there are three distinct phases in the record: (i) between 1885 and around 1930 concentration values peak for Al (12.28 ng g$^{-1}$), Ba (63.02 pg g$^{-1}$), Fe (2.07 ng g$^{-1}$), Mg (26.01 ng g$^{-1}$), S (29.66 ng g$^{-1}$), and Ti (69.54 pg g$^{-1}$); (ii) between





approximately 1930 and 1955 minimum concentrations can be observed for the above trace elements; and (iii) between
approximately 1955 and 2008 a second increasing trend is observed for Al, Ba, Fe, Mg, S, and Ti.

The measured trace element concentrations generally agree with previous studies (Kreutz et al., 2000b; Dixon et al., 2013)
but show visible variability during the 20[th] century. The ice core record reflects changes in emissions as well as atmospheric
circulation and transport processes. The El Niño-Southern Oscillation (ENSO) is the variation in southern Pacific sea surface
temperature that induces regional-scale changes in atmospheric circulation in those latitudes. Several researchers have
reported links between ENSO and aerosol deposition flux in Antarctica (Vance et al., 2013; Criscitiello et al., 2014). In the
section 3.4, we will discuss in detail the influence of climate variables in the concentration of trace elements presented.

Sea salt concentrations in snow and ice are influenced by sea ice concentration and wind speed in the source region. Some
studies show that processes associated with sea ice formation (e.g. frost flower formation, brine production, and blowing
snow released from sea ice surfaces) are the dominant source of sea salt aerosols for Antarctica (Rankin et al., 2002; Wolff et
al., 2003; Kaspari et al., 2005; Fischer et al., 2007; Criscitiello et al., 2013). Meanwhile, other authors suggest that strong
winds over open water are responsible for the introduction, transportation, and deposition of sea salt aerosols (Abram et al.,
2011; Udisti et al., 2012).

Winter concentration maxima are consistent with stronger winter wind speeds. On average, winter wind speeds measured by
automatic weather stations (AWS) at Byrd Station during 1980 to 2008 were 16.7 m s$^{-1}$, while only 9.3 m s$^{-1}$during summer
(SCAR READER project, available at https://legacy.bas.ac.uk/met/READER/data.html, accessed online in June 2016). This
is supported by Koffman et al. (2014) who found increased winter transport of mineral dust in a central WAIS ice core
record. Winter maxima of marine derived elements (e.g. Na, Sr, K) also suggest marine aerosols originated from the sea ice
surface in addition to the open ocean. Frost flowers that form on the surface of young sea ice are identified as an important
source of marine aerosols in high latitudes (Wolff et al., 2003; Kaspari et al., 2005).

While regional and local sources are important for sea salt aerosols, the main source regions for mineral dust are located
more than 4,000 km away (Krinner and Genthon, 2003). Ice-free regions correspond to only ~0.18 % of the Antarctica
continental area (Burton-Johnson et al., 2016). These regions can be an additional source of mineral dust input to the
bordering areas (Bory et al., 2010; Koffman et al., 2014), in particular those that are close to the Transantarctic Mountains
(Delmonte et al., 2013), Antarctica Peninsula (Bory et al., 2010), Ellsworth Mountains, Marie Byrd Land (Koffman et al.,
2014; Tuohy et al., 2015; Winton et al., 2016), and McMurdo Sound (Atkins and Dunbar, 2009; Dunbar et al., 2009;
Chewings et al., 2014). Mineral dust derived from remote continental sources and deposited in Antarctica typically consists
of particles smaller than <5 μm diameter (Delmonte et al., 2002; Gaiero et al., 2007), however, contribution from local
sources include particles >5 μm diameter (Mahowald et al., 2014). Coarser sized mineral dust particles were observed at
coastal sites (e.g. Roosevelt Island) and near the margin of the ice sheet (e.g. WAIS Divide) compared to distally sourced
dust deposited on the Antarctica Plateau (Koffman et al., 2014; Winton et al., 2016), suggesting input from local dust
sources. Previous work at the MJ coring site, Cataldo et al., (2013) found particle sizes ranging from 1.1 μm to 2.4 μm.





Based on these data we assume that the concentrations are more influenced by remote continental sources than local sources. Although we do not disregard that local contributions can be a secondary source for dust.

### 3.3 Atmospheric transport to Mount Johns ice core site

Atmospheric transport to Antarctica is dominated by the circumpolar westerly winds over the Southern Ocean and the permanent cyclone belt over the polar fronts (Hoskins and Hodges, 2005). The baroclinic zone, between 60-70°S, is a very active cyclone generating area due to the interaction of cold, dry air from the continent and relatively warmer, moist air from the Southern Ocean. The prevailing mid-latitude westerlies direct the cyclones, circulating around the Antarctic continent (King and Turner, 1997). The Antarctic plateau is dominated by a high-pressure (anticyclonic, counter-clockwise) and the

wind regime is governed by katabatic winds (cold, dense air flowing downhill due to gravity).

The Amundsen Sea Low (ASL) is a major driver of West Antarctic climate variability (Turner et al., 2013). It is a mobile, climatological low-pressure located between 170–298° E and 80–60° S, in the Southern Pacific (Kreutz et al., 2000b; Hosking et al., 2013). The depth and location of the low-pressure center affects the climatic conditions and the strength of the westerlies along the coastal regions and the interior of West Antarctica (Ding et al., 2011; Schneider et al., 2012).

Antarctic ice cores provide the data for reconstructing a record of aerosol transport to the continent where observations are scarce. It is important to identify and quantify the mineral dust and marine sources that influence a particular ice core site because it represents passive tracers, which can be used to reconstruct climate conditions in their source regions as well as large-scale atmospheric transport patterns. Previous studies have shown that mineral dust load to West Antarctica is tracked to the circum-Antarctic continents (Prospero et al., 2002; Li et al., 2010; Neff and Bertler, 2015) and ice-free regions within

Antarctica (Dixon et al., 2011; Albani et al., 2012b; Delmonte et al., 2013; Koffman et al., 2014). Back- trajectory modeling can be used to understand the synoptic controls on precipitation and the transport of trace elements to the study site. Here, we calculated 5-day, backward trajectories using the NOAA HySPLIT Model and clustered the trajectories over austral spring (September–November), summer (December–January), autumn (March–May), and winter (June–August) months. Daily simulations were generated during the 1979 to 2008 period. For the simulated trajectories, we have determined that

five clusters are sufficient to capture the seasonal trajectory variability.

The MJ site receives the majority of air masses from the Amundsen Sea and, secondarily, from across the Antarctic Peninsula and Weddell Sea (Fig. 5). An additional source could be a continental local contribution. It is possible to identify two clusters with dominant westerly flow patterns ranging from fast (long) to slow-moving (short) depending on the season; one cluster includes fast-moving trajectories with strong cyclonic curvature around the Ross Ice Shelf (whole year); a group

with direction that varies from westerly flow to northeasterly depending on the season, and a continental grouping that contains mainly katabatic flow paths from the interior. The clusters show that air masses circulate around the Antarctic continent until they are diverted to the interior, as synoptic storms, across the Amundsen and Bellingshausen Seas. In some cases, the air masses arrive on the WAIS after being diverted across the Weddell Sea and travelling over the Filchner-Ronne Ice Shelf.



The trajectories are classified into two groups: 1) oceanic influenced (blue), and 2) continental influenced (red). These classifications are defined by characteristics identified from the mean trajectories of each cluster, speed (proportional to trajectory length), source region, and pathway. The frequency distribution of cluster classes for the 1979–2008 period shows distinct seasonality between the austral summer and other seasons, with comparatively stronger, westerly transport in the cold months and a secondary, northeasterly transport in the warm months. On average, the oceanic group has a maximum,

seasonal frequency in winter while the continental group peaks in winter/spring. During the summer, the trajectories generally are slow-moving (short) and are more locally influenced than in others seasons.

The highest concentrations of trace elements in the winter concentrations are associated with air masses clustered within the oceanic trajectories. These air masses have the potential to capture impurities transported over the South Pacific from mid-latitude continental regions such as South America or Australia. Moreover, winds remobilize mineral dust from ice-free

areas in the WAIS, for example trajectories crossing Marie Byrd Land, Ellsworth Land, the Antarctic Peninsula, and the Weddell Sea. Li et al. (2008) show that due to the prevailing westerlies, the distribution and deposition of dust has an eastward transport so that half of the Atlantic Ocean and Indian Ocean are influenced by South American dust and the Pacific region is influenced by Australian dust. Modeling studies of dust transport to Antarctica (Krinner et al., 2010) show that the annual mean concentration of dust in West Antarctica (particularly Marie Byrd Land), is mostly represented by dust

originating from Australia. This is in agreement with Neff and Bertler (2015) and Tuohy et al. (2015) who show dust transportation from Australia and New Zealand to the South Pacific. As observed previously in other ice core sites from the WAIS (Dixon et al., 2011; Koffman et al., 2014; Neff and Bertler, 2015), mineral dust reaching the MJ ice core site is unlikely to be associated with a single dust source due to mixing along-transport from presumed continental sources.

**3.4 Interannual atmospheric variability**

Previous studies have shown that sea ice variability can be influenced by fluctuations in atmospheric circulation patterns associated with intraseasonal (e.g. Renwick et al. 2012), interannual (Kohyama and Hartmann, 2016), and decadal climate variability, such as the Southern Annular Mode (SAM) (Lefebvre et al., 2004; Stammerjohn et al., 2008), the El Niño Southern Oscillation (ENSO) (Simpkins et al., 2012), and the Atlantic Multidecadal Oscillation (AMO) (Li et al., 2014). Sea ice concentration trends are generally consistent with sea surface temperature (SST) trends, such that regions of increasing

(decreasing) sea ice are nearly always found in an environment of decreasing (increasing) SST. However, winds can influence sea ice concentration in several ways, including atmospheric thermal advection, oceanic currents, and wind-driven dynamic transport (Holland and Kwok, 2012). Schneider et al. (2011) suggest that atmospheric circulation trends have influenced both the sea ice concentration and the temperature trends in the Pacific sector of the Antarctic.

Correlations were made between annual means of reanalysis variables (SIC, SST, and 2-m air temperature) from the ERA-

Interim product and the studied trace element annual average concentrations from 1979–2008. Observed correlations suggest a relationship between sea ice concentration and aerosol transport to the ice core site. Figure 6 shows that correlations between SIC and most of the analyzed elements are positive ($r > 0.55$; $p < 0.05$) in the Ross Sea and negative ($r > -0.45$; $p <$





0.05) in the Bellingshausen Sea, consistent with others authors (Simpkins et al., 2012; Turner et al., 2015a). Previous studies using satellite observations show a dipole structure in SIC with increasing sea ice in the Ross Sea (Stammerjohn et al., 2015)

and decreasing in the Amundsen–Bellingshausen Seas (Holland and Kwok, 2012). This dipolar pattern is related to thermodynamic and dynamic forcing associated with variability in the pressure anomalies extending over the Amundsen Sea (Turner et al., 2015b).

The strongest positive correlations ($r = 0.65$; $p < 0.05$) are in the region between 180º and 140º W, suggesting that concentrations at the MJ site increase when SIC is high in the West Amundsen and Ross Seas. This may indicate the

dominant source of the marine aerosols in the MJ area. Aluminum and magnesium exhibit negative correlations ($r = -0.55$; $p < 0.05$) in the Ross Sea region and positive correlations ($r = 0.50$; $p < 0.05$) in the Antarctica Peninsula area. Aluminum in polar ice derives almost exclusively from crustal dust (McConnell et al., 2007), which explains the correlation difference between Al and other trace elements. However, Mg exhibits the same pattern observed for Al, showing a relationship between the two elements (which has been observed previously in the Pearson correlation). Based on this, we believe that

Mg has an important secondary source (mineral dust, in this case) in addition to the marine aerosols. Similar results were previously discussed in Pasteris et al. (2014).

SST plays an important role in oceanic heat content controlling the interactions between the ocean and atmosphere (Reynolds et al., 2007). SST variability in the Southern Hemisphere (SH), related to ENSO, is associated with changes in the atmospheric circulation (Ding et al., 2012; Fogt et al., 2015), Southern Ocean SST anomalies (Simpkins et al., 2014; Ciasto

et al., 2015), Antarctic SIC anomalies (Simpkins et al., 2012; Turner et al., 2015a), and Antarctic surface temperature anomalies (Schneider et al., 2012; Steig et al., 2015). Criscitiello et al. (2014) demonstrated that tropical Pacific SST anomalies have influenced the source and transport of marine aerosols to the WAIS.

Ding and others (2011) show that anomalous SST under areas of strong tropical convection in the central tropical Pacific have generated an atmospheric Rossby wave response that influences atmospheric circulation over the Amundsen Sea,

causing increased warm air advection to the WAIS. Other studies suggest that tropical SST forcing has a significant impact on the southern annular mode (SAM) during austral winter (Ding et al., 2012; Li et al., 2014), deepening the Amundsen–Bellingshausen Seas low (Li et al., 2015). Previous studies indicate that wind associated with low pressure systems over the Bellingshausen-Amundsen Sea would facilitate the generation and transport of sea-salt aerosols from either an open ocean or a sea ice source (Bromwich et al., 2013; Criscitiello et al., 2014; Pasteris et al., 2014), as well as influence dust transport

from circum-Antarctic continents to the Southern Ocean and WAIS (Neff and Bertler, 2015).

Figure 7 shows associations between SST and annual trace element concentrations. Negative correlations ($r > -0.70$; $p < 0.05$) between SST in the Ross, Amundsen, and Bellingshausen Seas with the trace element concentrations indicate increased transport of marine aerosols when the SST is cooler. Strong negative correlation ($r > -0.75$; $p < 0.05$) is also observed in the Weddell Sea and Indian Ocean. On the contrary, Al and Mg concentrations are positively correlated ($r >$

0.65; $p < 0.05$) with the Ross, Bellingshausen, and Amundsen Seas as well as the southern Indian Ocean SST indicating that Al and Mg concentrations increase when SST is high. Once more, Al and Mg exhibit a very strong relationship with each





other and an inverse correlation to that presented by the other elements. In the case of these two elements, transport and deposition processes seem to be associated with mineral dust transport and influenced by warmer SST between the source and the MJ site.

The 1979–2008 correlations for $T_{2m}$ are displayed in Fig. 8. Annual $T_{2m}$ anomaly maps generally reveal weak, negative anomalies ($r > -0.45$; $p < 0.05$) over the Ross and West Amundsen Seas for the measured elements, while showing low, positive anomalies ($r > 0.45$; $p < 0.05$) for Al and Mg specifically.

Ligtenberg et al., (2013) show that snowfall and $T_{2m}$ in the Antarctic continent appear to be linked: periods with increasing $T_{2m}$ coincide with periods of increasing snowfall. Changes in $T_{2m}$ show warmer and moister conditions extending

considerably farther inland, consistent with enhanced air intrusions (Nicolas and Bromwich 2011). Changes in snow accumulation are also linked to the deepening of the Amundsen Sea Low (ASL), tropical SST, and large-scale atmospheric circulation (Thomas et al., 2015).

Our results are consistent with other authors who have identified that seasonal concentration maxima in sea salt elements correlate well with the SIC winter maxima (Rankin et al., 2002; Sneed et al., 2011; Abram et al., 2013; Fan et al., 2014).

Seasonal and interannual variability of the trace element concentrations are also likely due to wind speed and transport efficiency at the time of deposition. Hoskins and Hodges (2005) suggest that storminess over the Southern Ocean and the strength of inland transport, both of which are enhanced during winter, can explain the winter maxima deposition in the WAIS.

The strong, negative correlation between trace element concentrations and SST further demonstrates that concentrations

contained in the MJ ice core record provide an indication of past variability. Previous studies have also identified a teleconnection between the tropical Pacific Ocean and aerosol deposition in Antarctica (Vance et al., 2013; Criscitiello et al., 2014). SST anomalies under areas of strong tropical convection have a significant influence on the atmospheric circulation in the Bellingshausen-Amundsen Sea area through the generation of a large-scale, atmospheric wave train (Lachlan-Cope and Connolley, 2006; Ding et al., 2011). The wave train pattern is prominent in winter and spring months (Lachlan-Cope and

Connolley, 2006). Since sea salt aerosols are deposited throughout the year, notably in winter, it is expected that tropical forcing and atmospheric Rossby waves would influence trace element concentrations.

Based on significant, negative correlations with SST, and supported by annual concentration variability of Ba, Fe, K, Mn, Na, and Sr, we suggest that high concentrations observed before 1930 in our ice core record are directly related to cooler SST affecting atmospheric transport in the Amundsen-Ross Sea region. We infer that the low concentrations observed later

(1930-1955) are consistent with a warming in the region. Schneider et al. (2008) show extreme positive anomalies (representative of West Antarctic surface temperature) during the 1936-1945 period. The authors interpreted these anomalies as indicative of strong teleconnections in part driven by the ENSO (1939–1942). Therefore, it seems possible that different trace element concentrations respond to different forcings on different timescales. Further studies are required to understand the influence of climate forcings on deposition patterns.



## 4. Conclusions

Using high-resolution ICP-SFMS, several trace elements were measured in Mount Johns ice core from WAIS and used to evaluate inputs from natural aerosol emission sources. It was found that natural contributions from mineral dust are important sources for Al and Ti, while marine aerosols from the open sea and sea ice in the southern Pacific are important sources of Na, Sr, and K, over the MJ site. However, Ba, Ca, Fe, Mg, Mn, and S show to be from mixed sources (mineral dust and sea-salt aerosols). Additionally, S and Mn exhibit important volcanic contribution and S has a considerable biogenic input in the summer season.

The MJ site receives the majority of air masses from the Amundsen Sea and, secondarily, from across the Antarctic Peninsula and Weddell Sea. Utilizing back trajectories (HySPLIT) were identified two dominant air-mass trajectory clusters: marine and continental. The frequency distribution of cluster classes for the 1979–2008 period, showed distinct seasonality between the austral summer and other seasons, with comparatively stronger, westerly transport in the cold months and a secondary, northeasterly transport in the warm months. On average, the oceanic group has a maximum, seasonal frequency in winter while the continental group peaks in winter/spring. During the summer, the trajectories generally are slow-moving (short) and are more locally influenced than in others seasons.

Finally, the reanalysis trace element correlations show that marine derived trace element concentrations are strongly influenced by SIC and SST anomalies. The results confirm that seasonal elemental concentration maxima in sea-salt elements occur during the sea ice concentration winter maxima in the West Amundsen and Ross Seas, as well as the transport of marine aerosols increases when SST is relatively cooler. The strong, negative correlation between trace element concentrations and SST further demonstrates that concentrations contained in the MJ ice core record can provide an indication of past variability. The high concentrations observed before 1930 in the ice core record are directly related to cooler SST affecting atmospheric transport in the Amundsen-Ross Sea region, as well as the low concentrations observed later (1930-1955) are consistent with a warming in the region.

We are demonstrating that both sources and transport of mineral dust and marine aerosols to West Antarctica are controlled by the climate variables in response to remote atmospheric forcing. While these results are promising, further work is needed to obtain a more detailed picture of past variability and its relationship with regional aerosol transport.

**Acknowledgments**

This research is part of the Brazilian Antarctic Program (PROANTAR) and was financed with funds from the National Council for Scientific and Technological Development (CNPq), project 407888/2013-6.

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




**Table 1. Statistical summary of Trace Elements Concentrations determined in MJ ice core.**

| Elements | Mean | Median | SD | Min | Max | MDL* |
|---|---|---|---|---|---|---|
| Al (ng g$^{-1}$) | 2.64 | 2.46 | 1.65 | 0.30 | 22.77 | 0.30 |
| Ba (pg g$^{-1}$) | 9.31 | 5.49 | 11.45 | 0.38 | 93.46 | 0.38 |
| Ca (ng g$^{-1}$) | 3.01 | 1.88 | 3.24 | 0.09 | 21.22 | 0.09 |
| Fe (ng g$^{-1}$) | 0.62 | 0.41 | 0.68 | 0.05 | 5.58 | 0.05 |
| K (ng g$^{-1}$) | 1.78 | 0.94 | 3.39 | 0.06 | 39.54 | 0.06 |
| Mg (ng g$^{-1}$) | 9.27 | 9.09 | 4.71 | 0.66 | 45.18 | 0.38 |
| Mn (pg g$^{-1}$) | 28.10 | 11.99 | 61.60 | 0.94 | 783.79 | 0.94 |
| Na (ng g$^{-1}$) | 21.91 | 12.64 | 33.10 | 0.21 | 381.61 | 0.21 |
| S (ng g$^{-1}$) | 10.24 | 9.01 | 6.30 | 0.58 | 62.53 | 0.06 |
| Sr (pg g$^{-1}$) | 23.21 | 17.74 | 17.85 | 0.74 | 117.94 | 0.74 |
| Ti (pg g$^{-1}$) | 14.15 | 9.00 | 18.76 | 0.66 | 209.35 | 0.66 |

*MDL = Method Detection Limit.

**Table 2. Global mean volcanic quiescent degassing background minimum and maximum and Mount Erebus volcanic plume minimum and maximum contributions at MJ site.**

| Elements | Concentration | Excess Concentration | Global Volcanic | | Local Volcanic | |
|---|---|---|---|---|---|---|
| | | | Max | Min | Max | Min |
| Al (ng g$^{-1}$) | 2.64 | < 0.01 | – | – | 0.0000025 | 0.0000015 |
| Ca (ng g$^{-1}$) | 3.01 | 1.24 | – | – | 0.0000019 | 0.0000012 |
| Fe (ng g$^{-1}$) | 0.62 | < 0.01 | – | – | 0.0000011 | 0.0000006 |
| K (ng g$^{-1}$) | 1.78 | 0.05 | – | – | 0.0000035 | 0.0000021 |
| Mn (pg g$^{-1}$) | 28.10 | 11.43 | 3.84 | 2.31 | 0.63 | 0.42 |
| Na (ng g$^{-1}$) | 21.91 | < 0.01 | – | – | 0.0000039 | 0.0000024 |
| S (ng g$^{-1}$) | 10.24 | 8.15 | 1.49 | 0.99 | 0.49 | 0.29 |
| Ti (pg g$^{-1}$) | 14.15 | < 0.01 | – | – | 0.0004 | 0.00024 |





**Table 3. Pearson's correlation coefficient determined for 2,137 samples in the MJ ice core. In bold type are the correlations with 99 % of confidence.**

|      | Al | Ba   | Fe   | Mn   | Ti   | Ca   | Mg   | Na   | Sr   | K    | S    |
|------|----|------|------|------|------|------|------|------|------|------|------|
| Al   | 1  | 0.34 | 0.23 | 0.26 | **0.40** | 0.10 | **0.75** | 0.15 | 0.21 | 0.15 | 0.33 |
| Ba   |    | 1    | **0.44** | **0.64** | 0.39 | 0.39 | 0.34 | **0.47** | **0.46** | **0.52** | **0.41** |
| Fe   |    |      | 1    | 0.29 | 0.38 | 0.20 | 0.10 | 0.18 | 0.21 | 0.25 | 0.17 |
| Mn   |    |      |      | 1    | 0.22 | 0.29 | **0.42** | **0.72** | **0.42** | **0.59** | **0.55** |
| Ti   |    |      |      |      | 1    | 0.14 | 0.25 | 0.12 | 0.24 | 0.13 | 0.27 |
| Ca   |    |      |      |      |      | 1    | 0.18 | 0.29 | **0.42** | 0.30 | 0.19 |
| Mg   |    |      |      |      |      |      | 1    | **0.51** | **0.58** | 0.29 | **0.40** |
| Na   |    |      |      |      |      |      |      | 1    | **0.75** | **0.64** | **0.41** |
| Sr   |    |      |      |      |      |      |      |      | 1    | **0.45** | 0.33 |
| K    |    |      |      |      |      |      |      |      |      | 1    | 0.34 |
| S    |    |      |      |      |      |      |      |      |      |      | 1    |




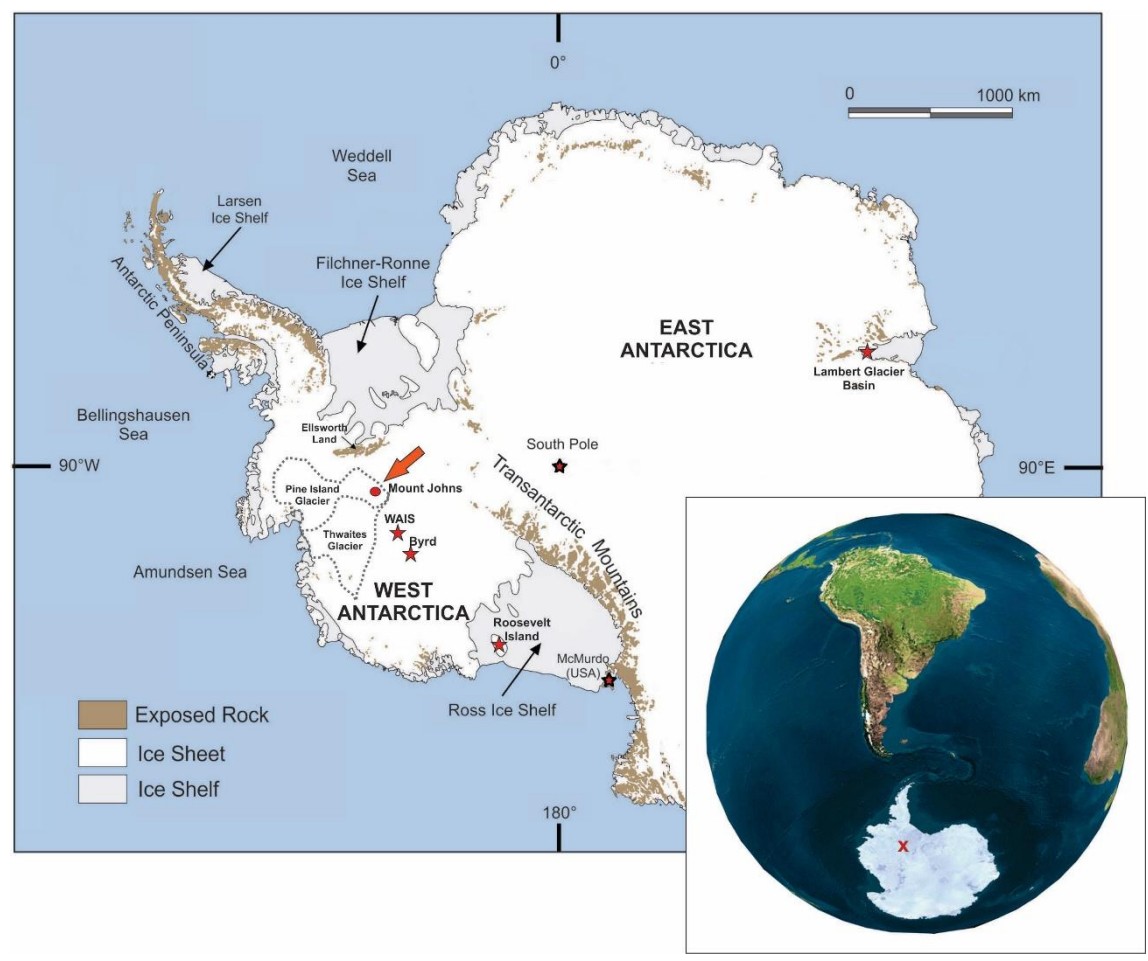

**Figure 1. Map of Antarctica showing the site of the MJ ice core site (red arrow) and locations of sites discussed in the text (figure adapted from the U.S. Geological Survey, http://lima.usgs.gov/).**





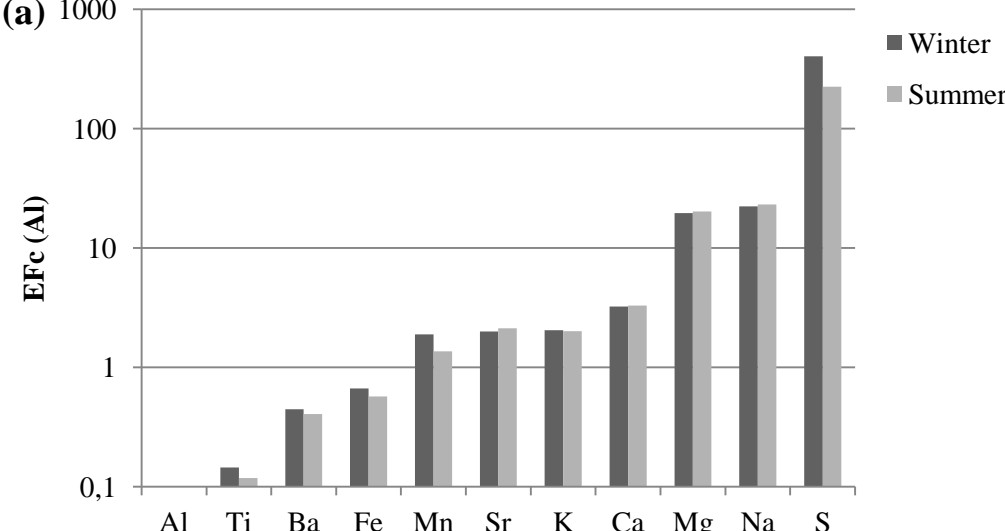

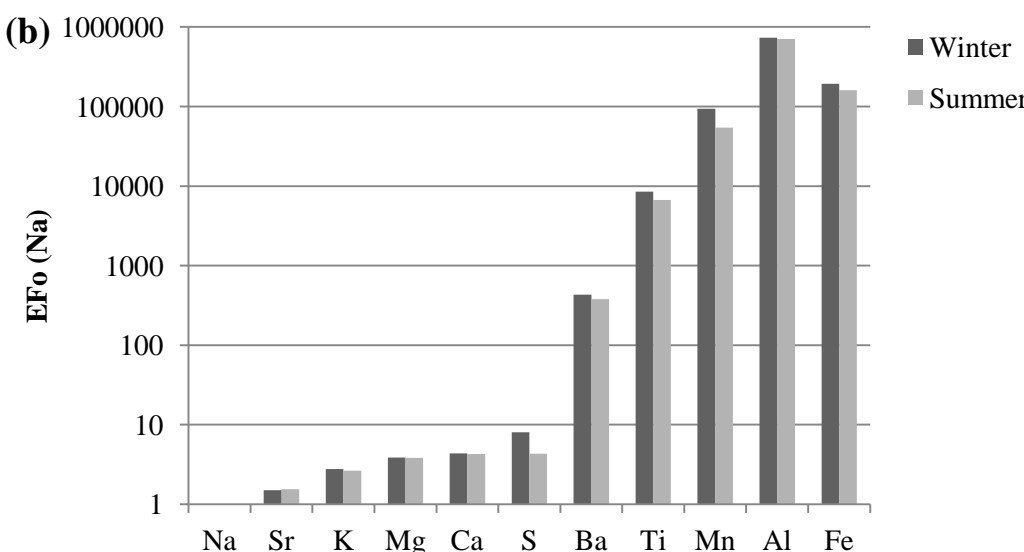

**Figure 2. (a) Mean element enrichment factors in reference to Earth's crust (EFc) and (b) oceanic composition (EFo) at the MJ site for austral summer (December to February) and austral winter (June to August).**




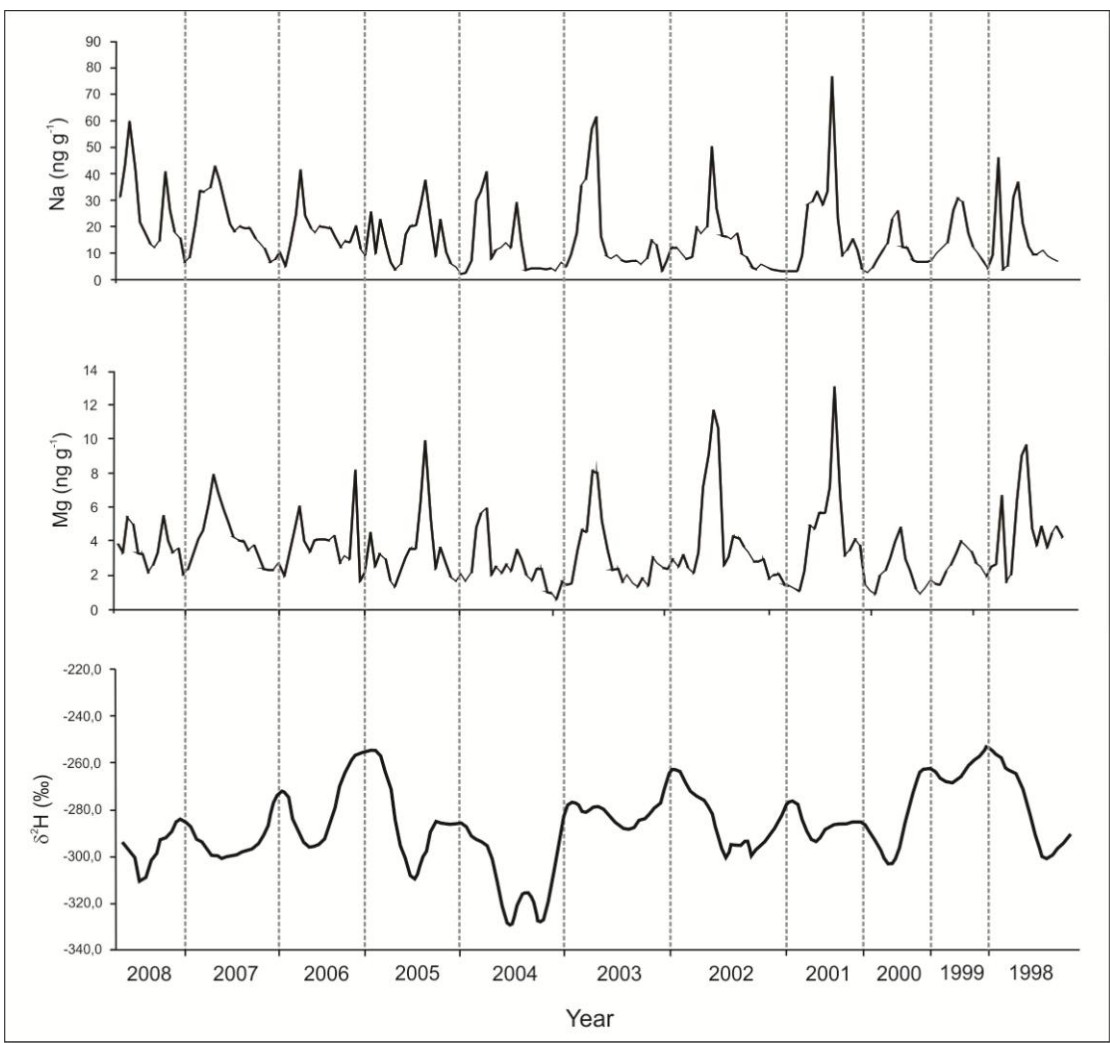

**Figure 3. Na, Mg, and $\delta^2$H variability concentrations measured in the MJ ice core over a range of 11 years. Na and Mg concentrations peaks mark winter layers.**







**Figure 4. Annual average concentrations of Al, Ba, Ca, Fe, K, Mg, Mn, Na, S, Sr, and Ti measured in the MJ ice core. Colored bands define three distinct phases in the record from 1883 to 2008. The peaks shaded in red indicate volcanic eruptions corresponding to events: Krakatau (1883), Santa Maria (1902), Agung (1963), and Pinatubo (1991).**





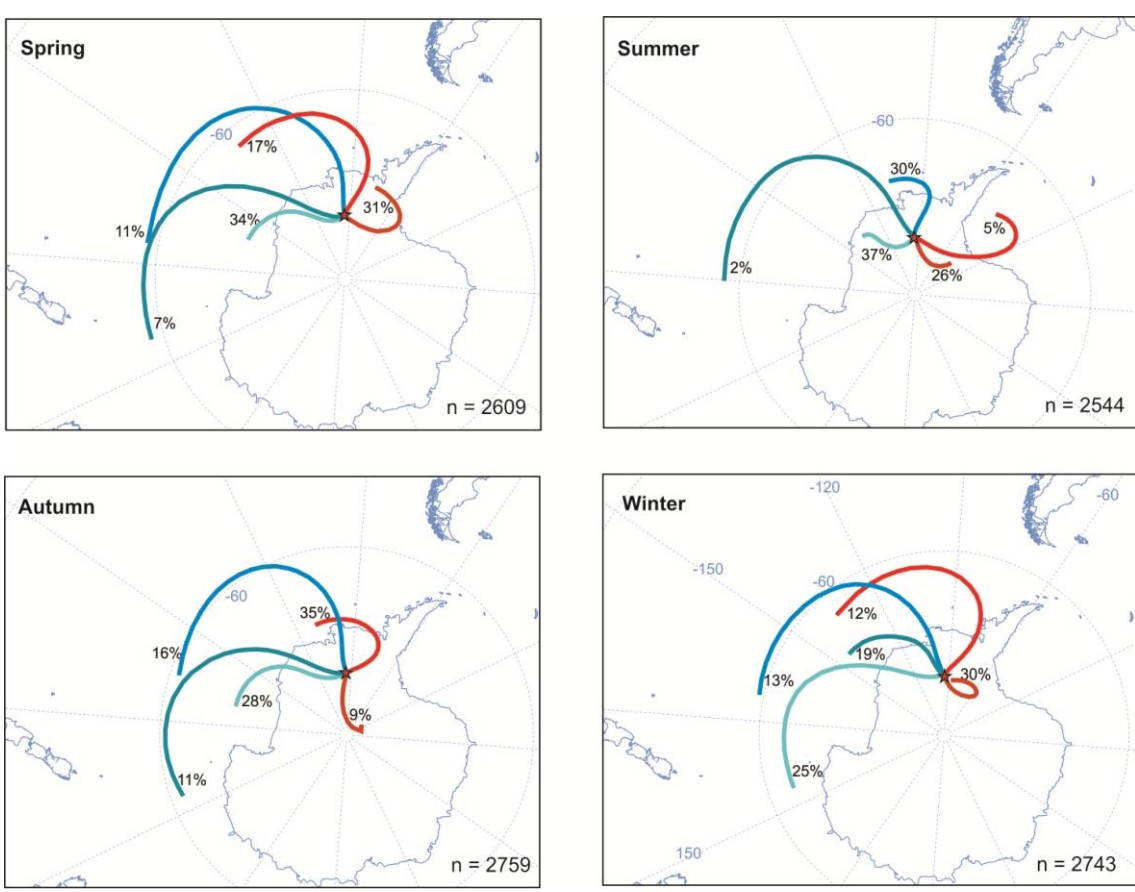

**Figure 5. HySPLIT seasonal clusters of daily 5 day back trajectories from 1979 to 2008 arriving at the Mount Johns ice core site, West Antarctica. Percentage of daily trajectories included in each cluster is indicated, number of daily trajectories for each season is indicated at the bottom right of each panel. Blue shaded area represent oceanic group clusters, while red shaded area show the continental group clusters. Trajectories calculated using the NOAA Hysplit Model (version 4.9).**



**Figure 6: Correlation of the 1979–2008 ERA-Interim reanalysis sea ice concentration versus annually-averaged trace element time series in the MJ ice core.**



**Figure 7: Correlation of the 1979–2008 ERA-Interim reanalysis sea surface temperature (SST) parameter versus the annually-averaged trace elements time series in the MJ ice core.**



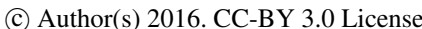

**Figure 8: Correlation of the 1979–2008 ERA-Interim reanalysis 2-m air temperature parameter versus the annually-averaged trace elements time series in the MJ ice core.**

810