# Peer review of "A 125-year record of climate and chemistry variability at the Pine Island Glacier ice divide, Antarctica"

_The Cryosphere, 2016_

## Referee Comment (RC1) · Anonymous Referee #1 · 28 Nov 2016

General Comments

The work from Schwanck et al. concerns the detailed analysis of a new West Antarctic ice core (92 m long). An accurate elemental characterization of the core is here presented, based on the determination of several elements with an impressive resolution. Chemical records from 1979 to 2008 are thus compared to meteorological data reanalysis and to atmospheric back-trajectories. Among the many points discussed in the work two are of extremely interest, making the work worth to be published in this journal. 1- an impressive work was carried out to obtain an elemental compositional database composed by 2137 samples. 2- the possibility to compare such database with meteorological data has a great and poorly unexplored potential for the understanding of aerosol transport and deposition in Antarctica. Coupling ice core records and meteorological observations will give important advances to the interpretation of data extracted from ice cores.

Despite the great potential I believe that the paper could be greatly improved with further and additional data analysis. If on one side the authors could realize such an impressive database, the discussion side of the paper is rather poor. Discussion should be generally improved and some additional statistical tools could be applied to extract further information from the data and to obtain robust evidences. Several points present a poor discussion. A new revised version of the paper will substantially benefit from further data treatment. My final suggestion is to consider the work for publication after major revision.

Specific Comments

The key point of the work is the comparison between meteorological and ice core data. This was possible because the considered site presents a high snow accumulation rate which allowed obtaining records with high temporal resolution. The description about the development and validation of the chronology is rather poor, not to say completely lacking. If high resolution meteorological data are compared to record obtained from natural archives, it is necessary for the latter ones to be accurately and precisely dated. The authors just state that the chronology was based on annual layer counting (using Na and S records) with the additional consideration of 4 major volcanic eruptions. No further details are given. A previous work (Schwank et al., 2016 Atmos. Environ.) is cited as reference for the chronology, but also in this work few details are found. This part needs a substantial extension. A first element would be the comparison between annual layer counting and historical eruptions, which error is found? Is this consistent with a record which is claimed to present a seasonal resolution?

Another important part of the paper is dedicated to the calculation of different contributions for each element, i.e. crustal, volcanic, marine and biogenic. This part is a little

bit confused. The authors follow three different approach: the selection of reference elements and reference elemental ratios, the calculation of enrichment factors and the calculation of Pearson's coefficients. It would be important to put all this elements together, discussing them in a comprehensive way and not separately. If the discussion is kept separated controversial results are found. For example we can consider Mg. According to the use of reference elements and ratios it has a dominant marine source (30 %, supplementary material) and a secondary associated to crustal material (5 %). But Pearson's coefficients reveal that Mg is strongly associated to Al, a typical crustal element. Also successive interpretation about the comparison with meteorological data point to strong similarities. The application of a multi-variate statistical tool as principal component analysis could greatly improve this section of the work. PCA could help the authors to identify different contributes and to understand the role played by each element in these different contributions. Since its starting point is the calculation of Pearson coefficients please consider to make a further step in this sense and complete data treatment with PCA. In addition I suggest the authors to improve the method they used to distinguish ss and n-ss Na. The assumption that Al is only crustal is justified, but this is not the case for the assumption that Na is only marine. Please consider to separate the two fractions by using Al as crustal reference and an UCC Al/Na ratio.

Section 3.2 should be deeply revised. In its current version it seems a review about atmospheric depositional issues in Antarctica, but very poor discussion points are reported. High time resolution data described in this work should be better exploited to understand seasonal dynamics. Are elements presenting parallel seasonal oscillations? If this is not the case and no significative observations are found please consider to dramatically shorten the section and to merge it with section 3.4, so as to have a single section about temporal variability.

Section 3.4 I suggest to develop the discussion presented here with a comparison with back-trajectories analysis and seasonal trends. Some interesting trends are observed but their interpretation is poor. The authors present a huge amount of observations

concerning literature and what was observed in other studies, but the connection between their evidences and literature is lacking. For example looking at Fig.7 the correlation of Al and Mg is completely different with respect to the other elements. This is clearly pointed in the text, but a true interpretation is missing. The phenomenon could be related to a different seasonality pattern, with dust peaks and marine aerosol peaks occurring in different periods of the year, when SST is different.

In the light of these comments conclusive remarks will need a final revision.

Technical Comments

Line16: insert "of" between "reanalysis" and "trace"

Line22: remove "that of the"

Line27: please consider to add a short passage about the importance of WAIS in relation to climate dynamics and sea level.

Line40: change "recognize" with "distinguish"

Line41-42: remove from "furthermore" to "continent" and replace with "both presenting specific seasonal cycles."

Line54-56: please reformulate, it is not clear.

Line56: change with "Another primary source of aerosol is mineral dust. It is transported..."

Line57: add a further reference. Li et al., 2008 is based only on models, add Revel-Rolland et al., 2006 EPSL which is based on isotopic data from EPICA Dome C. Also the consideration of New Zealand as dust source for Antarctica is still only an hypothesis based on modeling works, no direct evidences are known.

Line69: add "of WAIS" after "systems"

Line76: please give a reference for modern snow accumulation rates in the considered

area

Line86: in the text Mount Johns is never described. Is it a topographical height of Pine Glacier? A peripheral area of this glacial system?

Line116-117: please specify only significant digits

Line310: some references concern Talos Dome, which is located in EAIS, not WAIS

Table1: is it possible to add a further column with average uncertainty for each element?

Figure2: I guess that y-axis of upper figure is wrong. Al EF should be 1, not 0.1. Is this right?

Figure3: here you present some examples to show seasonal variations. You considered Na and Mg. What about considering also Al? Being exclusively crustal it could present a different behavior.

Figure4: specify in the caption that volcanic eruptions were identified using sulfates

Figure6-7-8: Why for each figure you report different elements. It would be nice to have three perfectly comparable figures with all the elements you considered in this work. Did you try to apply the same procedure to nss and ss-S. It would be nice to see them.

Best Regards

---

## Referee Comment (RC2) · Anonymous Referee #2 · 11 Dec 2016

In this manuscript the authors present elemental analysis from an ice core located on the Antarctic Peninsula. They combine this analysis with backtrack trajectory modeling, identifying the major air masses pathways to the site. They also correlate elemental time series with sea-ice thickness and SSTs around the continent, identifying regions that correlate with marine tracers. Once the very good and exhaustive literature review is subtracted, there is unfortunately not much substance to this manuscript, as it presents very few new insights. It appears to be made up of three sections, one about the elemental analysis and the correlation with SIC and SST, one second about marine vs crustal contribution of aerosol impurities, and one third about the backtrack trajectory modeling, and no link between the three. Because of this, the paper is mostly a

data paper presenting elemental concentrations and a theoretical exercise with Hysplit, and the combination of the two is not stronger that the separate halves. Although the theme of this manuscript fits within the scope of The Cryosphere, I think the method and the analysis are flawed to the point that the results will be significantly altered. For this reason I suggest to reject the paper at this point so that it can be completely rewritten with no time constraint and resubmitted at a later time.

Major comments:

1. ERA and NCEP data are notoriously unreliable over Antarctica, with huge biases compared to measurements. Especially the 1000m winds rely mostly on simulated model values, which are also notoriously wrong in Antarctica. Bracegirdle and Marshall (2012) may have determined that ERA-Interim data are the most accurate of the 6 reanalysis models, but that doesn't make them correct or even close to reality. What's the sensitivity of your results when using the other reanalysis datasets? How do the ERA data compare with climatology timeseries of monitoring stations close to MJ?

2. The whole chapter 3.1 is methodically flawed (see minor comments below). This puts into doubt most of the interpretation based on these data. See the minor comments below.

3. In chapter 3.3 you mention that modeling studies suggest Australia as the main source for the Antarctic Peninsula, but that ice core studies mostly identify a mix of sources. What about your results of non-marine tracers, you don't mention these in the paper. If you decide to concentrate on marine tracers, then the calculation of nss concentrations is not necessary.

4. You mention that in the Antarctic peninsula, wet deposition dominates and the concentration of trace elements depends on cyclonic activity, which is episodic and seasonally variable. However, you do all the correlation analysis using annual means and I don't think that's representative.

5. One of your conclusions is that "marine derived trace element concentrations are strongly influenced by sea ice concentration and sea surface temperature anomalies". That is a wrong conclusion, all you have is a correlation analysis, no dynamical or physical explanation to imply causation.

Minor comments:

Line 56: Mineral dust is not a source of aerosols.

Line 84: Location of SST and sea ice changes?

Line 98: Please always use SI units. In the case of clean rooms that would be the ISO 14644-1 standard. A class 100 room is equivalent to an ISO 5 level (10ˆ5 particles per cubic meter). The class 100, 1000, etc. standard has been obsolete for over 15 years, it's time people move on.

Line 115-119: That method makes no sense to me, although I may just be ignorant of this matter. The standard deviation of the measurements should have no relevance for the detection limit? The instrument could be very precise at medium range, but have a detection limit greater than it's precision. Or did you mean the average of the blanks plus 3 times the std?

Line 120-125: Where did the samples come from? Did you send frozen pieces of the ice core to Brazil? If so how were they treated in Brazil? Or did you send aliquots from the fraction collector? If so how did you send them? Frozen?

Line 127-129: Briefly mention here why S can be used to count layers. Why did you not use other measurements, such as Ca or Al, for the layer counting?

Line 148 – 152: Where is your dust source? If it's Oceania are 5 days enough to transport the particles across the Pacific?

Line 173: The regression line in Figure S1 is just ridiculous. Obviously there is no linear relationship between Na and nssS. Please use common sense and don't blindly

apply methods found in other papers.

Line 202-209: How can you distinguish your calculated excess from the error introduced by crustal Na and oceanic Al? I doubt anything below 10% contribution is significant, once you calculate the calculation uncertainty due to these effects.

Line 214-220: Have you looked at the distributions? Are the elements normally distributed? I doubt it and you cannot use Pearson's correlation then. Try the Spearman or Kendall correlation instead. And redo the classification of crustal and marine elements.

Line 226-227: It may be best to remove table S3 unless you can address all the comments above.

Line 253: Mean of what? And do you really have a 0.01 pg/g measuring accuracy? Please go through all the text and remove all those decimals.

Line 325-327: How exactly were these classification defined? It sound rather subjective to me, was there an objective criteria? What about South American influence?

Line 333-334: You don't need to cross the Pacific from South America. The South American contribution would come through the South Atlantic cluster.

Figures:

Figure 2: Have the same sequence of elements in both (a) and (b) plots.

Figure 3: remove "concentrations" after variability
* * *

---

## Editor Comment (EC1) · J. Savarino (Editor) · 23 Jan 2017

Dear Authors

I have read the interactive comments and I should warn you that at this stage I'm not convinced by your replies. Both reviewers draw serious critics about the data treatment and interpretations during the first stage of the review process. I did not find your replies convincing and they lack details. Most of the time, the replies refer and point to change made in the revised version but none of the reviewers have access to it, neither me at the time of writing.

I would like to see the change that you intend to do (or did in the revised version) right

in your replies to the reviewer's comments and not systematically pointing to a revised manuscript. Also I would like to see all questions and comments of the reviewers addressed in your replies. For instance, R2 asked What's the sensitivity of your results when using the other reanalysis datasets? Question that is not answered in your reply.

The interactive public discussion stage does not allow to look at a revised version. Therefore all anticipated changes should be clearly stated, with details in the reply to reviewer's comments.

Sincerely.
* * *

---

## Author Comment (AC3) · 31 Jan 2017

Dear Dr. J Savarino,

We are working hard to improve this paper, some changes still need to be made and we believe that before the end of February this paper could be considered ready. We are adding a new file with answers to the reviewer's comments. We apologize for the first file does not display all the details about the modifications that we are being made and we hope it is now as expected.

Sincerely,

Franciéle Schwanck

---

## Editor Comment (EC2) · J. Savarino (Editor) · 7 Feb 2017

Dear Franciele

please can you submit a revised version? I want the reviewers to read it and gave me their new assessments.

With regards

---

## Author Response (AR1)

We very much thank the two reviewers for their thorough analysis of our paper and for their valuable comments and suggestions. They had been carefully considered and most of them are accounted in the revised manuscript. Answers and explanations to all detailed questions and annotations raised by the reviewers are provided in the following.

(RC: Reviewer comments; AC: Author comments).

**Reviewer comments 1**

**Specific Comments**

**RC 1:** The description about the development and validation of the chronology is rather poor, not to say completely lacking. If high resolution meteorological data are compared to record obtained from natural archives, it is necessary for the latter ones to be accurately and precisely dated. The authors just state that the chronology was based on annual layer counting (using Na and S records) with the additional consideration of 4 major volcanic eruptions. No further details are given. A previous work (Schwank et al., 2016 Atmos. Environ.) is cited as reference for the chronology, but also in this work few details are found. This part needs a substantial extension. A first element would be the comparison between annual layer counting and historical eruptions, which error is found? Is this consistent with a record which is claimed to present a seasonal resolution?

**AC:** *The dating was improved with the use of stable isotope data (these data were not available until then). More details about dating will be added to the text and to the supplement information. Manual interpretation of the data was done by multiple individuals to identify the individual layers. The CCI software package (Kurbatov et al., 2005) was also used to identify matching seasonal peaks from Ca, Na and Sr and the major historical volcanic eruptions. In this study, water isotopes were used to confirm the dating previously performed in Schwanck et al., 2016.*

**RC 1:** Another important part of the paper is dedicated to the calculation of different contributions for each element, i.e. crustal, volcanic, marine and biogenic. This part is a little bit confused. The authors follow three different approach: the selection of reference elements and reference elemental ratios, the

calculation of enrichment factors and the calculation of Pearson's coefficients. It would be important to
put all this elements together, discussing them in a comprehensive way and not separately. If the
discussion is kept separated controversial results are found. For example, we can consider Mg.
According to the use of reference, elements and ratios it has a dominant marine source (30 %,
supplementary material) and a secondary associated to crustal material (5 %). But Pearson's coefficients
reveal that Mg is strongly associated to Al, a typical crustal element. Also successive interpretation
about the comparison with meteorological data point to strong similarities. The application of a multi-
variate statistical tool as principal component analysis could greatly improve this section of the work.
PCA could help the authors to identify different contributes and to understand the role played by each
element in these different contributions. Since its starting point is the calculation of Pearson coefficients
please consider to make a further step in this sense and complete data treatment with PCA.

**AC:** *We agreed that this chapter was confusing, so we decided to take the comment into account and
redo the analysis using PCA. The PCA resulted in four PCs. PC1 is dominated by Ba, K, Mg, Mn, Na,
and Sr, accounting for 42.24% of the total variance. PC2, dominated by Al and Ti, accounts for 13.27%
of the total variance, while K and Na are negatively correlated. PC3 is dominated by Ba, Fe, and Ti,
accounting for 11.16% of the total variance. PC4 is dominated by Ca and Sr, accounting for 8.11% of
the total variance, while S and Mn are negatively correlated. We are still working on the interpretation
of these results.*

**RC 1:** In addition I suggest the authors to improve the method they used to distinguish ss and n-ss Na.
The assumption that Al is only crustal is justified, but this is not the case for the assumption that Na is
only marine. Please consider to separate the two fractions by using Al as crustal reference and an UCC
Al/Na ratio.

**AC:** *We revised the whole calculation of ss and nss taking Al into account as crustal reference.*
*Non-sea-salt ratios were calculated using the equation reported below (Palmer et al., 2002, Becagli et
al., 2005):*

$nssS = S - 0.084 \times ssNa,$

*where S is the total sulfur concentration on the sample, 0.084 is the mean S/Na ratio in seawater (Lide, 2005) and ssNa is the Na actually derived from sea spray. Since some Na derives from continental dust, ssNa was calculated using the four-equation system reported below:*

$$ssNa = Na - nssNa$$

$$nssNa = nssAl \times (Na/Al)_{crust}$$

60 $$nssAl = Al - ssAl$$

$$ssAl = ssNa \times (Al/Na)_{seawater},$$

*where the mean Na/Al ratio is 0.3315 in the crust (Wedepohl, 1995) and the mean Al/Na ratio is 0.000000185 in seawater (Lide, 2005).*

65 **RC 1:** Section 3.2 should be deeply revised. In its current version it seems a review about atmospheric depositional issues in Antarctica, but very poor discussion points are reported. High time resolution data described in this work should be better exploited to understand seasonal dynamics. Are elements presenting parallel seasonal oscillations? If this is not the case and no significative observations are found please consider to dramatically shorten the section and to merge it with section 3.4, so as to have

70 a single section about temporal variability.

**AC:** *We agree. Sections 3.2 and 3.4 have been merged into a single section (Section 3.2 - Interannual atmospheric variability). We changed the order of discussion, placing the transport session (3.3 - Atmospheric transport to Mount Johns ice core site) to the end.*

*We are also expanded the discussion of results.*

75

**RC 1:** Section 3.4 I suggest to develop the discussion presented here with a comparison with back-trajectories analysis and seasonal trends. Some interesting trends are observed but their interpretation is poor. The authors present a huge amount of observations concerning literature and what was observed in other studies, but the connection between their evidences and literature is lacking. For example

80 looking at Fig.7 the correlation of Al and Mg is completely different with respect to the other elements. This is clearly pointed in the text, but a true interpretation is missing. The phenomenon could be related

to a different seasonality pattern, with dust peaks and marine aerosol peaks occurring in different periods of the year, when SST is different.

**AC:** *The section is being rewritten and improved. We agreed that some interpretations were superficial and we are expanding on these issues. The order of the sessions was changed, leaving the discussion of transport to the end. We are also expanding the discussion on how transport and seasonality have affected trace elements concentrations on MJ area.*

**Technical Comments**

**RC 1:** Line16: insert "of" between "reanalysis" and "trace"

**AC:** *Done*

**RC 1:** Line22: remove "that of the"

**AC:** *Done*

**RC 1:** Line27: please consider to add a short passage about the importance of WAIS in relation to climate dynamics and sea level.

**AC:** *The text was changed to: "During recent decades, rapid changes have occurred in the WAIS sector, including flow velocity acceleration, retraction of ice streams, and mass loss (Pritchard et al., 2012). These changes influence the global climate through their contributions to sea level rise (Pritchard et al. 2009, Shepherd et al. 2012) and deep ocean circulations (Holland and Kwok, 2012). WAIS contains sufficient water to raise the global sea level by over 3 m (Bamber et al., 2009, Fretwell et al., 2013)".*

**RC 1:** Line40: change "recognize" with "distinguish"

**AC:** *Done*

**RC 1:** Line41-42: remove from "furthermore" to "continent" and replace with "both presenting specific seasonal cycles."

110  **AC:** *The text was changed to: "To interpret chemistry records from Antarctic ice cores, it is imperative to distinguish the long-range transportation of continental dust and regionally derived sea salt, both presenting specific seasonal cycles."*

**RC 1:** Line54-56: please reformulate, it is not clear.

115  **AC:** *The text was changed to: "Marine aerosol concentrations are strongly linked to cyclone frequency and intensity that provides high wind speeds over the ocean surface, with the aerosols deposited along the storm track."*

**RC 1:** Line56: change with "Another primary source of aerosol is mineral dust. It is

120  transported. . ."
**AC:** *Done*

**RC 1:** Line57: add a further reference. Li et al., 2008 is based only on models, add Revel-Rolland et al., 2006 EPSL which is based on isotopic data from EPICA Dome C. Also the consideration of New

125  Zealand as dust source for Antarctica is still only an hypothesis based on modeling works, no direct evidences are known.
**AC:** *Done*

**RC 1:** Line69: add "of WAIS" after "systems"

130  **AC:** *Done*

**RC 1:** Line76: please give a reference for modern snow accumulation rates in the considered area
**AC:** *We add Medley et al., 2014*

135  **RC 1:** Line86: In the text, Mount Johns is never described. Is it a topographical height of Pine Glacier? A peripheral area of this glacial system?
**AC:** *Mount Johns is a nunatak in the Pine Island Glacier area.*

**RC 1:** Line116-117: please specify only significant digits

**AC:** *We have removed the values. Details about MDL values are contained in the supplement information.*

**RC 1:** Line310: some references concern Talos Dome, which is located in EAIS, not WAIS.

**AC:** *The reference was being used as an example but to avoid misunderstanding we decided to remove.*

**RC 1:** Table1: is it possible to add a further column with average uncertainty for each element?

**AC:** *Done*

**RC 1:** Figure2: I guess that y-axis of upper figure is wrong. Al EF should be 1, not 0.1. Is this right?

**AC:** *This is correct; some elements presented EFc less than 1.*

**RC 1:** Figure3: here you present some examples to show seasonal variations. You considered Na and Mg. What about considering also Al? Being exclusively crustal it could present a different behavior.

**AC:** *Figure 3 was removed from the text and added to the supplement information. Mg was replaced by Al in the graph.*

**RC 1:** Figure4: specify in the caption that volcanic eruptions were identified using sulfates.

**AC:** *Done*

**RC 1:** Figure6-7-8: Why for each figure you report different elements. It would be nice to have three perfectly comparable figures with all the elements you considered in this work. Did you try to apply the same procedure to nss and ss-S. It would be nice to see them.

**AC:** *The graphs that were not shown in the figure did not present very significant results. These have now been added. We did not simulate for ss-S and nss-S only for total S.*

**Reviewer comments 2**

**Major comments:**

**RC2:** ERA and NCEP data are notoriously unreliable over Antarctica, with huge biases compared to measurements. Especially the 1000m winds rely mostly on simulated model values, which are also notoriously wrong in Antarctica. Bracegirdle and Marshall (2012) may have determined that ERA-Interim data are the most accurate of the 6 reanalysis models, but that doesn't make them correct or even close to reality. What's the sensitivity of your results when using the other reanalysis datasets? How do the ERA data compare with climatology time series of monitoring stations close to MJ?

**AC:** *The observations in Polar Regions, such as Antarctica, are extremely scarce making reanalysis heavily simulated in these regions. Biases are to be expected where observations are scarce. Since the closest weather observation site is Byrd Station (over 550 km away and about 550 m lower in elevation), even a comparison between reanalysis models and Byrd Station would not give an accurate estimate of how reanalysis compare to atmospheric conditions at Mount Johns. We understand that biases, whether large or small, are to be expected over Mount Johns due to a lack of weather observations; however, reanalysis models are the best estimate of atmospheric states in this region and is necessary for studies such as this. Because there are several climate reanalysis datasets available for investigating climatological behavior we inter-compare our results between ERA-Interim and an ensemble average of the four leading third-generation reanalyzes models (Gen 3) (Auger et al., in review). The models within Gen 3 are CFSR (Climate Forecast System Reanalysis), MERRA (Modern Era Retrospective Reanalysis for Research and Applications), JRA55 (Japanese 55-year Reanalysis), and ERAI. The variables behavior are captured equally well in both ERAI and Gen 3 ensemble representations. Therefore, we considered that the sensitivity of the correlation results on this work using different datasets is low.*

- *Auger, J.D., Birkel, S.D., Maasch, K.A., Mayewski, P.A., Schuenemann, K.C., 2017. An ensemble average and evaluation of third generation global climate reanalysis models. J. Geophys. Res. (in review).*

**RC2:** The whole chapter 3.1 is methodically flawed (see minor comments below). This puts into doubt most of the interpretation based on these data.

**AC:** The chapter has been rewritten and improved. We revised the whole calculation of ss and nss considering Al as crustal reference (details below). In addition, we replaced the Pearson's correlation by Principal Component Analysis.

**RC2:** In chapter 3.3, you mention that modeling studies suggest Australia as the main source for the Antarctic Peninsula, but that ice core studies mostly identify a mix of sources. What about your results of non-marine tracers, you don't mention these in the paper. If you decide to concentrate on marine tracers, then the calculation of nss concentrations is not necessary.

**AC:** *In this chapter we focus on atmospheric transport through trajectory simulations, the model does not allow us to directly use the measured concentrations. However, we made associations between the trajectories and the studied elements looking for marine and continental influences on the concentrations. Some modifications were made in the text, making clearer the discussion about nss-elements.*

**RC2:** You mention that in the Antarctic Peninsula, wet deposition dominates and the concentration of trace elements depends on cyclonic activity, which is episodic and seasonally variable. However, you do all the correlation analysis using annual means and I don't think that's representative.

**AC:** *It is difficult to separate the element concentrations into seasons. Although we discuss seasonality of elemental concentrations, it is mainly summer and winter, or low and high, respectively. Therefore, we decided that the best way to make these correlations is using annual concentrations to correlate with annual means of atmospheric variables.*

**RC2:** One of your conclusions is that "marine derived trace element concentrations are strongly influenced by sea ice concentration and sea surface temperature anomalies". That is a wrong conclusion; all you have is a correlation analysis, no dynamical or physical explanation to imply causation.

**AC:** *We do not consider this erroneous conclusion, since we use various tools (trajectory simulations, correlations with atmospheric variables, and statistical analyzes) that show us the relations between*

*elements of marine origin and sea ice and also the influence of temperature on concentrations (which is*
*clear in winter concentrations).*

225

**Minor comments:**

**RC2:** Line 56: Mineral dust is not a source of aerosols.

**AC:** *The sentence was rewritten to: "Another primary source of trace elements is mineral dust"*

230    **RC2:** Line 84: Location of SST and sea ice changes?

**AC:** *The sentence was rewritten to: "Regional changes in atmospheric circulation and associated*
*changes in tropical Pacific sea surface temperature and sea ice extent also directly influence the*
*warming trend in West Antarctica"*

235    **RC2:** Line 98: Please always use SI units. In the case of clean rooms that would be the ISO 14644-1
standard. A class 100 room is equivalent to an ISO 5 level ($10^5$ particles per cubic meter). The class
100, 1000, etc. standard has been obsolete for over 15 years, it's time people move on.

**AC:** *We have researched recent references and all refer to class-100 or class-1000. We decided to keep*
*it as in the original.*

240    - *Uglietti et al. 2015. Widespread pollution of the South American atmosphere predates the*
     *industrial revolution by 240 y. PNAS, v. 112(8), p. 2349-2354.*
    - *Tuohy, A. et al. 2015. Transport and deposition of heavy metals in the Ross Sea Region,*
     *Antarctica, Journal of Geophysical Research: Atmospheres, v. 120 (20), p. 10996-11011.*

245    **RC2:** Line 115-119: That method makes no sense to me, although I may just be ignorant of this matter.
The standard deviation of the measurements should have no relevance for the detection limit? The
instrument could be very precise at medium range, but have a detection limit greater than it's precision.
Or did you mean the average of the blanks plus 3 times the std?

**AC:** *The instrument is very precise. Blanks are made to control possible contaminations during the*
250  *melting process and the acidification. Considering that the blanks (ultrapure water) are coming from*

*different steps of the analytical procedure, it can be considered the real limiting factor for the determination of trace elements at the low and sub pg/g level. The detection limits were defined as three times the standard deviation of blank samples (10 blank samples were used). Concentrations below the detection limits were disregarded. This occurred in less than 1% of the samples.*

255 *Reference used:*

- *Barbante et al. 1997. Direct determination of heavy metals at pictogram per gram levels in Greenland and Antarctic snow by double focusing inductively coupled plasma mass spectrometry. Journal of analytical atomic spectrometry, v.12, p. 925-931)*

260 **RC2:** Line 120-125: Where did the samples come from? Did you send frozen pieces of the ice core to Brazil? If so how were they treated in Brazil? Or did you send aliquots from the fraction collector? If so how did you send them? Frozen?

**AC:** *The samples collected at the CCI were sent frozen to Brazil and melted on the day of analysis.*

265 **RC2:** Line 127-129: Briefly mention here why S can be used to count layers. Why did you not use other measurements, such as Ca or Al, for the layer counting?

**AC:** *The dating was improved with the use of stable isotope data (these data were not available until then). More details about dating will be added to the text and to the supplement information. Manual interpretation of the data was done by multiple individuals to identify the individual layers. The CCI*
270 *software package (Kurbatov et al., 2005) was also used to identify matching seasonal peaks from Ca, Na and Sr and the major historical volcanic eruptions. In this study, water isotopes were used to confirm the dating previously performed in Schwanck et al., 2016.*

**RC2:** Line 148 – 152: Where is your dust source? If it's Oceania are 5 days enough to
275 transport the particles across the Pacific?

**AC:** *We believe MJ presents a mixture of sources with main contribution from Australia followed by South America. The five day simulation is an appropriate time-length when considering the maximum*

*lifetime transport (10 days) of small size (0.1 − 2.5 μm) fractions of mineral dust and other aerosols, while transport of large particles (> 2.5 μm) is likely restricted to the first few days.*

280

**RC2:** Line 173: The regression line in Figure S1 is just ridiculous. Obviously there is no linear relationship between Na and nssS. Please use common sense and don't blindly apply methods found in other papers.

**AC:** *This part has been removed. Due to less than 1% of the samples being affected by sulfur*
285 *fractionation we decided not to apply the correction.*

**RC2:** Line 202-209: How can you distinguish your calculated excess from the error introduced by crustal Na and oceanic Al? I doubt anything below 10% contribution is significant, once you calculate the calculation uncertainty due to these effects.

290 **AC:** *We only use ssNa for the calculation and the oceanic contribution for the aluminum is so low that we consider insignificant. See calculations of nss and ss added.*

*Non-sea-salt ratios were calculated using the equation reported below (Palmer et al., 2002, Becagli et al., 2005):*

$$nssS = S - 0.084 \times ssNa,$$

295 *where S is the total sulfur concentration on the sample, 0.084 is the mean S/Na ratio in seawater (Lide, 2005) and ssNa is the Na actually derived from sea spray. Since some Na derives from continental dust, ssNa was calculated using the four-equation system reported below:*

$$ssNa = Na - nssNa$$

$$nssNa = nssAl \times (Na/Al)_{crust}$$

$$nssAl = Al - ssAl$$

300 $$ssAl = ssNa \times (Al/Na)_{seawater,}$$

*where the mean Na/Al ratio is 0.3315 in the crust (Wedepohl, 1995) and the mean Al/Na ratio is 0.000000185 in seawater (Lide, 2005).*

**RC2:** Line 214-220: Have you looked at the distributions? Are the elements normally distributed? I doubt it and you cannot use Pearson's correlation then. Try the Spearman or Kendall correlation instead. And redo the classification of crustal and marine elements.

**AC:** *No, the elements do not present normal distribution. We improved the analysis using Principal Component Analysis. The PCA resulted in four PCs. PC1 is dominated by Ba, K, Mg, Mn, Na, and Sr, accounting for 42.24% of the total variance. PC2, dominated by Al and Ti, accounts for 13.27% of the total variance, while K and Na are negatively correlated. PC3 is dominated by Ba, Fe, and Ti, accounting for 11.16% of the total variance. PC4 is dominated by Ca and Sr, accounting for 8.11% of the total variance, while S and Mn are negatively correlated. We are still working on the classification of these results.*

**RC2:** Line 226-227: It may be best to remove table S3 unless you can address all the comments above.

**AC:** *Due to the changes we made in the text, we decided that Table S3 was no longer needed and was removed.*

**RC2:** Line 253: Mean of what? And do you really have a 0.01 pg/g measuring accuracy? Please go through all the text and remove all those decimals.

**AC:** *Yes. Our analyses have this accuracy.*

**RC2:** Line 325-327: How exactly were these classification defined? It sound rather subjective to me, was there an objective criteria? What about South American influence?

**AC:** *These classifications are defined by characteristics identified from the mean trajectories of each cluster, speed (proportional to trajectory length), source region, and pathway.*

**RC2:** Line 333-334: You don't need to cross the Pacific from South America. The South American contribution would come through the South Atlantic cluster.

**AC:** *Yes. We agreed.*

**RC2:** Figure 2: Have the same sequence of elements in both (a) and (b) plots.

**AC:** *The elements are now in the same sequence.*

335 **RC2:** Figure 3: remove "concentrations" after variability

**AC:** *Figure 3 was removed from the text and added to the supplement information. Mg was replaced by Al in the graph.*

[revised manuscript text omitted]
 resulted in four PCs reporting 74.77% of the total variance of 11 chemical variables (Table 2). PC1 is dominated by Ba, K, Mg, Mn, Na, and Sr, accounting for 42.24% of the total variance. PC2, dominated by Al and Ti, and with K and Na negatively correlated accounts for 13.27% of the total variance. PC3 is dominated by Fe, accounting for 11.16% of the total variance, while Mg is negatively correlated. Finally, PC4 is dominated by Ca and Sr, accounting for 8.11% of the total variance, while S and Mn are negatively correlated.

Cluster analysis (CA) was performed to further classify elements of different sources based on the similarities of their chemical properties. In this study was used Ward's method, with Euclidean distances as the criterion for forming clusters of elements. Figure 3 displays four clusters: (1) Al–Mg–S; (2) Ba–Mn–Ti; (3) Fe–K; (4) Ca–Na–Sr. It is observed, however, that clusters 2 and 3 join together at a relatively higher level implying perhaps a common source, while the distance between Al–Mg and S in cluster 1 may suggest that this cluster can be further divided into two sub clusters.

Based on the analysis of crustal and marine enrichment factors, PCA and CA, we have classified the concentrations as predominantly crustal for the elements Al and Mg, while Na, Sr, and Ca are primarily sea salt derived elements. The elements Ba, Mn, Ti, Fe, and K show to be from mixed sources (mineral dust and sea salt aerosol). Furthermore, the S record has a considerable volcanic and biogenic input and Mn has an additional volcanic input. We acknowledge that Fe may have

an additional contribution of biomass burning aerosol (Winton et al., 2016), but due to lack of data we will not address this issue here. Based on the low values of EFc and EFo, we consider that the presented concentrations are from natural origin and possible anthropogenic contributions to these elements would be insignificant in this area.

585  ## 3.2 Intraseasonal and Interannual atmospheric variability

Generally, trace element concentrations from sea salt aerosol observed in coastal and interior West Antarctic ice cores show a clear seasonal signal, with higher concentrations in austral winter and spring months (June to November) and lower concentrations in austral summer months (December-February) (Legrand and Mayewski, 1997; Wolff et al., 2003; Kaspari et al., 2005; Sigl et al., 2016). Impurities from continental dust can peak in both the austral summer (Weller et al., 2008; Tuohy

590  et al., 2015) or winter months (Hur et al., 2007), depending on site location. Additionally, biogenic aerosols (e.g. sulfur) show peaks in summer months due to an increased phytoplankton activity (Weller et al., 2011). We found high concentrations in austral winter and low concentrations in austral summer for most of the elements analyzed, with the exception of sulfur, which presents peaks in the summer due to the biogenic contribution and of the elements with crustal influence (Al, Ba, Mg, and Ti) that present the highest concentrations in spring/summer. This variability was confirmed by

595  the seasonal cycle of water isotopes. The $\delta^2$H maximum represents annual summer peaks, while the $\delta^2$H 
[revised manuscript text omitted]

[Figure]

Figure 3. Cluster analysis calculated from the principal components score matrix extracted from the entire data set. Trace elements on PC1 are divided into two subgroups composed of Ba, Mn, Ti, Fe, and K (mixed sources) and Ca, Na, and Sr (marine component). Cluster analysis adds PC2, a subgroup composed of Al and Mg (crustal component) and PC4 composed of S (volcanic influence).

1140

[Figure]

Figure 4. Annual average concentrations of Al, Ba, Ca, Fe, K, Mg, Mn, Na, S, Sr, and Ti measured in the MJ ice core. Colored bands define three distinct phases in the record from 1883 to 2008. The peaks shaded in red were identified using sulfur record and indicate volcanic eruptions corresponding to events: Krakatau (1883), Santa Maria (1902), Agung (1963), and Pinatubo (1991).

1145

1150

[Figure]

**Figure 5: Correlation of the 1979–2008 ERA-Interim reanalysis sea ice concentration versus annually-averaged trace element time series in the MJ ice core.**

1155

[Figure]

**Figure 6: Correlation of the 1979–2008 ERA-Interim reanalysis sea surface temperature (SST) parameter versus the annually-averaged trace elements time series in the MJ ice core.**

[Figure]

1160

**Figure 7: Correlation of the 1979–2008 ERA-Interim reanalysis 2-m air temperature parameter versus the annually-averaged trace elements time series in the MJ ice core.**

[Figure]

1165 **Figure 8. HySPLIT seasonal clusters of daily 5 day back trajectories from 1979 to 2008 arriving at the Mount Johns ice core site, West Antarctica. Percentage of daily trajectories included in each cluster is indicated, number of daily trajectories for each season is indicated at the bottom right of each panel. Blue shaded area represent oceanic group clusters, while red shaded area show the continental group clusters. Trajectories calculated using the NOAA Hysplit Model (version 4.9).**

1170

---

## Author Response (AR2)

We very much thank the reviewer for your valuable comments and suggestions. Answers and explanations to all detailed questions and annotations raised by the reviewer is provided in the following. **Marked-up manuscript version was add at the end.**

(RC: Reviewer comments; AC: Author comments).

**Reviewer comments**

**Specific Comments**

**RC:** I found this new version of the manuscript greatly improved with respect to the first one. But I think that further work is needed. The data presented in this work are so abundant and rich that in order to prepare a balanced paper also a great effort in terms of interpretation and data-analysis are required. This is the weak side of the manuscript. But I am sure that with a little bit more work it will be possible to draw interesting conclusions and evidences. In general, I suggest to the authors to shorten a lot of sections. Reading the manuscript sometimes it seems to read a phd-thesis, not a draft of a paper. The space given to references and to their explanations is simply too much. Please consider to resume the most important information taken from literature in a more concise way, not presenting each single reference with an entire sentence, which describe it. This is particularly true for chapter 3.2, but also other sections of the work are similar.

**AC:** *We tried to simplify the text by removing some parts. We also split the text into more chapters making reading easier.*

*New chapters:*

*3.1 Dating of the ice core*

*3.2 Glaciochemical records*

*3.3 Principal component analysis*

*3.4 Intraseasonal concentration fluxes variability*

*3.5 Interannual atmospheric variability*

*3.6 Atmospheric transport to Mount Johns ice core site*

*3.7 Relationships between atmospheric circulation, temperature, and sea ice concentration*

**RC:** Moving to the content I have some doubts about the interpretation. What is lacking is a comprehensive analysis of all the data. At the present state, the discussion about elemental fractions, PCA and cluster is completely separated from back trajectories

analysis and from the meteorological considerations. TO draw robust conclusions putting all the pieces together is necessary.

**AC:** *We added a new chapter, "3.7 Relationships between atmospheric circulation, temperature, and sea ice concentration", where all results are discussed together and in a clearer way.*

**RC:** Some details about the analysis of chemical data. At first the equation system presented to distinguish the ss and nss fractions of Al and Na is not convincing. In the equations, you show that nss Na depends on nss Al and that ss Al depends on ss Na. Without setting, a reference how is it possible to solve this system? Sodium is a function of Aluminum and vice versa, I really could't understand. You need to take an assumption if you want to solve the problem. I think that the only reasonable possible one is to assume that Al is only crustal and that nss Al is null. Assuming this point it will be possible to distinguish Na ss and nss fractions.

**AC:** *Yes, we were assuming Al as crustal origin only; the sentence was rewritten making it clear. The text was changed to: "In this study, we assume that Al is only from crustal origin (total Al is equal to nssAl)."*

**RC:** About PCA. I don't really agree with your interpretation of PC. In addition I think that with such good results their interpretation should be deepened. My idea about PCA is:

-PC1 is a sort of impurity index. We see that each element is positively correlated with this PC. This means that when the atmospheric load is high (high impurity content) PC1 show high values. At the opposite when impurity concentration reaches minima values PC1 is low. It would be interesting to show the record of PC1 along depth (or age) and see if presents seasonal values, I guess so.

-PC2 and PC3 are related to dust vs marine aerosol deposition. Here we see that Ba, Fe, Ti, Al are positively correlated, while other elements related to marine aerosol as K and Na are negatively correlated. These components distinguish the two fractions. Probably if we look further in deep we could say that PC2 highlights aluminosilicate (Al is the most important element), while PC3 highlights the siliclastic fraction of mineral dust (Fe and Ti).

-PC4 seems related to carbonate content, being Ca and Sr major constituents of CaCo3. I suggest to the authors to better develop this part, studying the trends related to each

component along the core and trying to understand the different seasonal cycles. This could help also for the successive part dedicated to the analysis of atmospheric transport and pathways.

**AC:** *We agree. The section is being rewritten and improved. We agreed that some interpretations were superficial and we are expanding on these issues.*

**RC:** Another important problem concerns the use of elemental concentration instead of fluxes. Since this is a well-dated core, with a good seasonal signal, I think that it would be worth to convert all concentrations in fluxes. In this way, it would be possible to highlight the real changes of deposition, excluding the effect associated to snow dilution. What is said here is that 3 major periods are recognized in terms of deposition, related to high and low concentration. Maybe such an effect is only related to changes of snow precipitation. Using fluxes such problems would be completely excluded and final interpretation would be improved.

**AC:** *Both concentration and flux presents similar variability, this indicates that the concentrations are independent of accumulation rate and that the variability exhibited in the record is likely not a function of changes in snow accumulation. Anyway, now we are presented the fluxes instead elemental concentrations, in the text.*

**Technical Comments**

**RC:** 102 and 108: class100, add ISO5
**AC:** *Done*

**RC:** 110: change to This process is important to guarantee a complete dissolution of particulate and non-soluble elemental fractions.
**AC:** *Done*

**RC:** 119: applied to, not as
**AC:** *Done*

**RC:** 122: (such event occurred in less than 1% of the samples); minor, not less
**AC:** *Done*

**RC:** 180-182: Al is not chosen because of its abundance. Al is chosen because in addition to being abundant (and easy to determine) is also quite stable and immobile from a chemical perspective. Thus, it can be reliably used as crustal tracer. (See for example nesbitt and young 1982)

**AC:** *The text was changed to: "Aluminum was used as the reference element in this work because it is a good proxy of continental dust"*

**RC:** 190: see my comments about Na and Al

**AC:** *Done*

**RC:** 217: I don't understand which those percentages are related to. In Tab S3 it seems that global volcanic deposition is more important than the local one, but in the text you state the opposite. Is it wrong or am I missing something?

**AC:** *The values were in reverse order. My mistake.*

**RC:** 249: specify that while the peak of species associated to marine aerosol is mainly related to winter sea ice peak, the peak of dust deposition is related mainly to atmospheric circulation patterns. In coastal sites such dynamics are greatly influenced by local factors, for this reason the dust peak doesn't occur in the same season in whole Antarctica, each site presents its peculiarities.

**AC:** *The section is being rewritten and improved.*

**RC:** 259: associated to

**AC:** *Done*

[revised manuscript text omitted]